# Transport of regional pollutants through a remote trans-Himalayan valley in Nepal

Shradda Dhungel[1], Bhogendra Kathayat[2], Khadak Mahata[3], Arnico Panday[1,4]

[1] Department of Environmental Sciences, University of Virginia, Charlottesville, VA 22904, USA
5   [2] Nepal Wireless, Shanti Marg, Pokhara, 33700, Nepal
[3] Institute for Advanced Sustainability Studies, Potsdam, 14467, Germany
[4] International Center for Integrated Mountain Development, Khulmaltar, Kathmandu, 44700, Nepal

*Correspondence to: Shradda Dhungel (shradda@virginia.edu)*

10   **Abstract.** Anthropogenic emissions from the combustion of fossil fuels and biomass in Asia have increased in recent years. High concentrations of reactive trace gases and light-absorbing and light-scattering particles from these sources form persistent haze layers, also known as atmospheric brown clouds, over the Indo-Gangetic Plain (IGP) from December through early June. Models and satellite imagery suggest that strong wind systems within deep Himalayan valleys are major pathways by which pollutants from the IGP are transported to the higher Himalaya. However, observational evidence of the transport of polluted air masses through Himalayan valleys has been lacking to date. To evaluate this pathway, we measured black carbon (BC), ozone ($O_3$), and associated meteorological conditions within the Kali-Gandaki Valley (KGV), Nepal, from January 2013 to July 2015. BC and $O_3$ varied over both diurnal and seasonal cycles. Relative to nighttime, mean BC and $O_3$ concentrations within the valley were higher during daytime when the up-valley flow (average velocity of 17 m s$^{-1}$) dominated. BC and $O_3$ concentrations also varied seasonally with minima during the monsoon season (July to September). Concentrations of both species subsequently increased post-monsoon and peaked during March to May. Average concentrations for $O_3$ during the seasonally representative months of April, August, and November were 41.7 ppbv, 24.5 ppbv, and 29.4 ppbv, respectively, while the corresponding BC concentrations were 1.17 µg m$^{-3}$, 0.24 µg m$^{-3}$, and 1.01 µg m$^{-3}$, respectively. Up-valley fluxes of BC were significantly greater than down-valley fluxes during all seasons. In addition, frequent episodes of BC concentrations two to three times higher than average persisted from several days to a week during non-monsoon months. Our observations of increases in BC concentration and fluxes in the valley, particularly during pre-monsoon, provide evidence that trans-Himalayan valleys are important conduits for transport of pollutants from the IGP to the higher Himalaya.

*Keywords: black carbon, ozone, trans-Himalayan valleys, pollutant pathways, long-range transport, regional transport episodes, short-lived climate forcers.*

30

## 1. Introduction

Persistent atmospheric haze, often referred to as Atmospheric Brown Cloud (ABC) (Ramanathan and Crutzen, 2003), affects broad geographic regions including the Indo-Gangetic plain (IGP) in southern Asia (Ramanathan and Carmichael, 2008), eastern China (Ma et al., 2010), southeast Asia (Engling and Gelencser, 2010), sub-Saharan Africa (Piketh et al., 1999), Mexico (Vasilyev et al., 1995), and Brazil (Kaufman et al., 1998). In southern Asia, the haze covers extensive areas particularly during the period of mid-November to mid-June preceding the summer monsoon season. Major combustion sources – including the anthropogenic burning of agricultural waste, garbage, biofuel, and fossil fuels as well as wildfires – emit volatile and particulate-phase compounds to the atmosphere that contain oxidized and reduced forms of sulfur, nitrogen, and organic carbon (OC) together with elemental (black) carbon (BC) and other species. These emissions are intermixed and chemically interact with mechanically produced aerosols (e.g., sea salt and mineral dust). Important secondary pollutants such as ozone ($O_3$) also form from photochemical reactions involving nitrogen oxides. Together, this mixture of atmospheric species constitutes the ABC or brown haze over South Asia (Ramanathan et al., 2005; Gustafsson et al., 2009). These optically thick layers include high concentrations of light-absorbing and light-scattering particles (Menon et al., 2002) that modulate radiative transfer.  Light-absorbing aerosols (primarily BC and crustal dust) contribute to atmospheric warming while light-scattering aerosols (primarily S-, N-, and OC-dominated particles) drive cooling at the surface. The combined effects of light-absorbing and light-scattering aerosols from anthropogenic sources serve to reduce UV and visible wavelength radiation at the surface (i.e., surface forcing), increase tropospheric warming (i.e., atmospheric forcing), and change the net top of the atmosphere solar flux (i.e., top-of-the-atmosphere forcing) (Andreae and Crutzen, 1997; Kaufman et al., 2002, Ramanathan et al., 2005). Light-absorbing anthropogenic pollutants like BC also significantly influence global warming, in terms of direct radiative forcing (Jacobson, 2001; Bond et al., 2013), and regional influences from such pollutants close to sources are greater than those on the global scale (Ramanathan et al., 2007b).

The elevated concentrations of aerosols in the anti-cyclone have also been shown to weaken circulation patterns and reduce total monsoon precipitation over southern India (Ramanathan et al, 2005; Fadnavis et al., 2013) while intensifying the monsoon over the foothills of the Himalaya (Lau et al., 2006). In addition to warming the atmosphere, the rising concentrations of BC and $O_3$ over southern Asia (e.g., Ramanathan and Carmichael, 2008) have detrimental impacts on human health. Recent work has shown that elevated levels of these two key pollutants can compromise cardiopulmonary and respiratory health (Krupnick et. al, 1990; Janssen et al., 2011). In addition, $O_3$ is a leading pollutant contributing to biodiversity loss (Royal Society, 2008) as well as declines in crop yields (Auffhammer et al., 2006).

Haze over the IGP often reaches heights of more than 3 km above sea level via convection and advection, and the Himalaya range forms a 2500 km long, 8 km high complex topographic barrier along the northern edge of the IGP (Singh et al., 2004; Dey and Di Girolamo, 2010; Gautam et al., 2011, Lüthi et al., 2015). Numerous studies have investigated the transport of pollutants from the IGP to the Himalayan foothills, the immediate source region for potential transport to the Tibetan Plateau (TP) (Pant et al., 2006; Dumka et al., 2008; Komppula et al., 2009; Hyvärinen et al., 2009; Ram et al., 2010; Brun et al.,

2011; Gautam et al., 2011; Srivastava et al., 2012). Further, a suite of studies involving satellite imagery (Ramanathan et al.,
2007a, Brun et al., 2011), back trajectories (Lu et al., 2011), model calculations (Kopacz et al., 2011; Zhang et al., 2015), ice
core analyses (Lee et al., 2008, Kang et al., 2010), and measurements in the higher Himalaya (Bonasoni et al., 2010, Chen et
al., 2017) strongly suggest that pollutants are efficiently transported from the IGP to the higher Himalaya and onto the
Tibetan Plateau, especially during spring prior to the monsoon. This transport north of the Himalayas is potentially
concerning as the TP plays a vital role in regulating the regional climate due to its effect on the Asian summer monsoon
(ASM) and the hydrologic cycle. In the TP, light-absorbing aerosols may not only serve warm the atmosphere but − when
deposited onto snow and ice surfaces − may also decrease albedo and thereby substantially increase the melting rates of
snow and glaciers (Kang et al., 2010). Indeed, model simulations (Qian et al., 2011) have shown that these light-absorbing
aerosols change the surface radiative flux in the higher Himalaya and the TP by 5 to 25 W m$^{-2}$ during the pre-monsoon
months of April and May. The interrelated perturbations of the ABC on radiative transfer, air quality, the hydrologic cycle,
and crop yields have important long-term implications for human health, food security, and economic activity over southern
Asia. However, to date, there has been little observational research to directly demonstrate the role of mountain valleys in the
transport of air pollutants from the IGP and the Himalayan foothills to higher elevations within and north of the Himalayan
range.

In other studied mountain ranges, it is well known that flow patterns carry polluted air masses up valleys to higher elevations
by providing a path of least resistance between tall mountains. In the European Alps, prevailing wind systems in the
mountain river valleys funnel polluted air from peripheral source regions to high elevations in a phenomenon known as
"Alpine Pumping" (Weissmann et al., 2005). Under fair weather conditions during daytime, the upslope winds are capable of
transporting significant pollutants and moisture into the free troposphere (Henne et al., 2004). Relative to air over plains, the
air within the valley heats and cools more quickly (Steinacker, 1984). The resultant differences in temperature create
gradients in pressure and density, which in turn drive the transport of air from the plains to higher elevations during the
daytime (Reiter and Tang 1984, Whiteman and Bian, 1998; Egger et al., 2000). Such direct evidence of a Himalayan
mountain valley wind system and its role in pollution transport has been observed at 5079 m a. s. l.., particularly during non-
monsoon seasons, in the Khumbu valley in Eastern Himalaya (Bonasoni et al., 2010). However, the transport patterns of
pollutants at the habitable mid-altitudes within a trans-Himalayan valley have yet to be observed. Here we present 2.5 years
of measurements of BC, O$_3$, and associated meteorological data from one of the deepest trans-Himalayan valleys, the Kali-
Gandaki Valley. We examine seasonal and diurnal patterns of BC and O$_3$, investigate potential episodes of enhanced
pollution transport up-valley, and make a preliminary estimate of BC mass transport. In doing so, we seek to provide the first
observational evidence of trans-Himalayan valleys acting as conduits for pollution transport from the IGP to the higher
Himalaya.

**2. Measurement Sites and Methods**

**2.1 Measurement Sites and Instrumentation**

This paper presents data from an atmospheric measurement station in Jomsom (28.87° N, 83.73° E, 2900 m asl) within the core region of the KGV along with four other automated weather stations up and down the valley from Jomsom (Figure 1). With the exception of aerosol optical depth measured as part of AERONET (AErosol RObotic NETwork) (Xu et al., 2015)

and a [14]C study at several sites in the Himalaya and TP, including Jomsom, (Li et al., 2016) no pollution data from Jomsom have previously been reported. Here we report diurnal and seasonal trends in two important SLCPs – BC and $O_3$ – to evaluate the role of trans-Himalayan valleys as pathways for the transport of polluted air from the IGP to the higher Himalaya.

The KGV is located in the Dhaulagiri zone of western Nepal (Fig. 1(a)). The KGV floor changes elevation from

approximately 1100 m to 4000 m above sea level (asl) over a horizontal distance of 90 km (Fig. 1(b)). Passing between the two eight thousand meter peaks of Dhaulagiri and Annapurna, it forms one of the deepest valleys in the world. The valley is a narrow gorge at the lower end and opens up into a wider, arid basin (Fig. 1(b)) with a maximum width of approximately one kilometer. The orientation of the KGV varies from the entrance to the exit. The general orientation of the valley is from SW (the mouth of the valley) to NE (the head of the valley) (Fig. 1). Approximately 13,000 inhabitants in several small

settlements sparsely populate the valley. Emission sources within the valley include biofuel combustion for cooking and fossil-fuel combustion by off-road vehicles. There is an airport in Jomsom which operates only in the morning, as the narrow valley width and high up-valley wind speeds make it dangerous for small aircraft to land at other times during the day. A total of 5245 vehicles have been registered in Dhaulagiri Zone since 2008, but most are based in the southern towns of Kusma, Baglung, and Beni, below 1 km altitude, at the lowest left corner of the map in Figure 1(b) (DOTM, 2016).

The atmospheric observatory at Jomsom (JSM_STA) is equipped with instruments to measure BC, $O_3$, and meteorology (Supplementary Table 1). The observatory is located on the southeast corner of a plateau jutting out from an east-facing slope about 100 m above the valley floor and with no major obstructions either up or down the valley. Equivalent black carbon (hereafter referred to as BC) was measured with a Thermo Multiangle Absorption photometer (MAAP), model 5012 that uses a multi-angle photometer to analyze the modification of radiation fields – as caused by deposited particles that

entered through a straight, vertical inlet line – in the forward and back hemisphere of a glass-fiber filter (GF-10). MAAP was operated at a flow rate of 20 L min$^{-1}$ and measured BC at a 1-minute frequency. We note that Hyvärinen (2013) illustrates the artifact in MAAP measurements in environments with high aerosol loading with an underestimation of concentrations above 9 µg m$^{-3}$. Since the median monthly concentrations for the duration of the measurement were less than 1µg m$^{-3}$ and 90$^{th}$ percentile measurements were below 2 µg m$^{-3}$ (and therefore below this threshold), MAAP corrections were not applied.

$O_3$ was measured with a 2B Tech model 205 via the attenuation of ultraviolet light at 254 nm passing through a 15 cm long absorption cell fitted with a quartz windows. The instrument was operated at a flow rate of 1.8 L min$^{-1}$. For instrument calibration, the BC instrument performed an automatic span and zero checks every 24 hours, while zero checks on the $O_3$ instrument were performed every 7 days. Wind speed and direction were measured by an automated weather station (JSM_2) installed on a ridge 800 m above the sampling site for BC and $O_3$ (JSM_1).

## 2.2 Data summary

The observatory operated from January 2013 through July 2015, but periodic power disruptions caused occasional data gaps (Figure 2). Unless otherwise noted, data reported herein correspond to periods when BC, $O_3$, and meteorological data were available simultaneously. Data were binned by season as follows: monsoon (July-September), post-monsoon (October-February) and pre-monsoon (March-June). Time of day corresponds to Nepal's local time zone (LT) (UTC + 5.75 h). From March 2015 to May 2015, four additional automated weather stations (10m in height) were operated along a longitudinal transect of the valley floor where wind speeds are typically the highest: near the entrance of the valley at Lete (LET), within the core at Marpha (MPH) and Jomsom (JSM_2), and near the valley exit at Eklobhatti (EKL) (Figure 1b). Power outages, instrument malfunctions, and a major earthquake in Nepal on April 25th, 2015 (and its aftershocks) limited the duration of records at all sites. However, between 1st and 14th May, all stations operated simultaneously, and the resulting data were used to evaluate the diurnal variability of wind fields along the valley.

BC, $O_3$, and meteorological data were averaged over 10 minute intervals. Up-valley (southwesterly) flows are defined as between 215° and 235° while down-valley (northeasterly) flows include data between 35° and 55°. Data for all days for which complete data were available over entire 24-hour periods were binned by season. The statistical significance of differences between up-valley and down-valley flow conditions during different seasons were evaluated using the non-parametric Kruskal Wallis and Mann-Whitney tests.

To normalize for the influence of day-to-day variability in absolute concentrations, relative diurnal variability in $O_3$ and BC concentrations measured during a given month were normalized to a common scale ranging from 0 to 1 (see e.g., Sander et al., 2003; Fischer et al., 2006). Normalized BC values for month $m$ were calculated as:

$$BC_{n,t} = \frac{BC_t - \min BC_m}{\max BC_m - \min BC_m}$$

where $BC_t$ is the BC concentration at time $t$ in the month $m$, and $\max BC_m$ and $\min BC_m$ are the maximum and minimum BC concentrations observed during month $m$. The normalized data were then binned into twenty-four, hourly increments. These calculations were repeated for $O_3$.

## 2.3 Preliminary flux estimates

Up-valley and down-valley BC fluxes were calculated separately and determined using wind direction measurements. Instantaneous flux at time $t$ was calculated as:

$$j_t = BC_t ws_t$$

where $BC_t$ is the BC concentration (mg BC m$^{-3}$) at time $t$ and $ws_t$ is the wind speed (m s$^{-1}$) at time $t$. From this, net daily mass

transport ($M_d$) for day $d$ was estimated as:

$$M_d = A \left( \sum \left( j_{t,up} \, \Delta t \right) - \sum \left( j_{t,down} \, \Delta t \right) \right)$$

where $A$ is the cross-sectional area of the valley at Jomsom ($1.41 \times 10^6$ m$^2$), $j_{t,up}$ and $j_{t,down}$ are the instantaneous up-valley and down-valley fluxes, respectively, occurring during day $d$, and $\Delta t$ is the length of each time step in seconds (i.e., 600 seconds). The cross-sectional area was estimated as a trapezoid with a height of 800 m (the difference in elevation between the two Jomsom stations) and cross-valley distances of 800 m (at JSM_1) and 2720 m (at JSM_2). The average of these two cross-valley distances (1760 m) multiplied by the height (800 m) yields the cross-sectional area of $1.41 \times 10^6$ m$^2$. Thus, net daily flux was calculated as the difference between the summation of instantaneous up-valley fluxes for day $d$ and the summation of instantaneous down-valley fluxes for day $d$.

If we assume that (1) the polluted boundary layer within the valley at Jomsom is 800 m deep (i.e., the approximate elevational difference between the two AWS sites at Jomsom), (2) BC within the polluted boundary layer is well mixed, and (3) wind velocities do not vary significantly with altitude through the polluted layer, the mass flux of BC through a vertical plane across the valley can be estimated. Supplementary Figure 1 shows that JSM_2 is well within the polluted haze layer during daytime/upvalley flows and that some BC is almost certainly transported above 800 m elevation above the valley floor. In addition, it is important to note that – by assuming a constant polluted boundary layer depth throughout the day – to a certain degree we likely underestimate up-valley flux during daytime and overestimate down-valley flux during nighttime. We therefore ensure that our estimates are conservative. The long atmospheric lifetime of particulate BC (several days to a week or more) – coupled with turbulent flow within the valley – supports the assumption that BC is well mixed. Supplementary figure 2 shows that the potential temperature (theta) gradient is less than zero between JSM_1 and JSM_2 which illustrates that the air within the valley is unstable. In addition, Egger et al. (2000) used theodolite measurements to demonstrate uniform wind speeds within the bottom 1000 m above the KGV floor at Jomsom and other locations. While we observed differences in wind speed of ~5 m s$^{-1}$ between the two Jomsom stations during the limited times when data were available from both, we were unable to determine whether this pattern persisted throughout the year. These data limitations also prevented us from a more in-depth assessment of potential nighttime decoupling. For these reasons, our only option was to follow the findings of Egger et al. (2000) and assign the wind speed at JSM_2 to the entire flux plane.

## 3. Results and Discussion

### 3.1 Evolution of local wind systems in the KGV

An understanding of the local wind regime is essential for analyzing pollution transport through mountain valleys. Measurements from JSM_2 show the diurnal evolution of wind at Jomsom in each season (Figure 3, Supplementary Figure 4). At JSM_2, up-valley flows are southwesterly and dominant during daytime, with peak velocities above 15 m s$^{-1}$ between 0900 LT to 1800 LT. Wind velocities decreased substantially after 1800 LT, with variable wind direction until midnight,

followed by northeasterly winds (during pre- and post-monsoon seasons (Figure 3 and Supplementary Figure 4b)). The wind patterns during monsoon appear strongly influenced by the monsoon anticyclone, an observation which is in agreement with wind direction measurements from other Himalayan valleys (Bonasoni et al., 2010; Ueno et al., 2008) (Supplementary Figure 4a). Although wind velocities at JSM_2 varied over the year, non-monsoon months exhibited similar diurnal patterns as a function of sunrise and sunset (Figure 3, and Supplementary Figure 4). As discussed below, this alternating pattern in

wind direction from strong daytime flows to weak nighttime flows during dry months resulted in a net transport of pollutants up the valley.

Our measurements at the four AWS stations on the valley floor illustrated the evolution of surface wind velocities along the length of the KGV (Supplementary Figure 3). In general, wind speeds along the valley floor were strongest within the core of the valley at MPH, JSM_1 and JSM_2 and were weaker in the entrance (LET) and exit (EKL) regions (Supplementary

Figure 3). The observation of these strong daytime wind speeds within the valley during is consistent with the hypothesis that wind patterns are modulated by a pressure gradient created from the differential heating of the arid valley floor relative to the mouth of the valley (Egger et al, 2000). In addition, our comparison of measurements at JSM_1 and JSM_2 (Figure 3; Supplementary Figures 3, 4a, and 4b) provided information regarding vertical variability in wind speed. Velocities at the higher elevation site of JSM_2 were about 5 m s$^{-1}$ and 3 m s$^{-1}$ greater than those near the valley floor during daytime and

nighttime, respectively. The two sites exhibited similar diurnal cycles with the exception of a relatively stronger northeasterly wind at JSM_2 from 0300 to 0900 LST.

### 3.2 Diurnal variability in BC and O$_3$

Based on median values, O$_3$ peaked during daytime and dropped to minimal levels before sunrise during all three study seasons (pre-monsoon, monsoon, and post-monsoon). However, in April 2013 (pre-monsoon period), O$_3$ peaked in the late

afternoon whereas in November 2014 (post-monsoon), it peaked in the early afternoon. In addition, the normalized diurnal excursions were greater during the pre- and post-monsoon periods relative to the monsoon period – represented by August 2014. In contrast, BC concentrations increased rapidly in the early morning, decreased during late morning, and then rose again during the afternoon and early evening hours (Figure 4); this pattern was consistent across all seasons, with relative diurnal variability being somewhat greater during post-monsoon relative to the pre-monsoon and monsoon periods. Across

all seasons, the lower normalized distributions for BC relative to O$_3$ reflect infrequent periods of high BC concentrations.

Several factors likely contributed to differences in timing of the daily peaks in O$_3$ and BC concentrations. These include potentially different source regions for BC and O$_3$ precursors, the timing of the photochemical production and destruction of O$_3$, and contributions of O$_3$ and its precursors from non-combustion sources like stratospheric ozone and biogenic hydrocarbons from vegetation. The early morning BC peak during all seasons suggests contributions from the local

combustion of biofuels for cooking and heating, which are most prevalent during early morning. The secondary peak in the afternoon and early evening occurred when the local anthropogenic emission sources were at a minimum in the KGV.

### 3.3 Seasonal variability in BC and O₃

All data generated during the measurement period were binned by month to evaluate the seasonal patterns of BC and $O_3$ (Figure 5). In addition, individual months with the most complete data coverage during the pre-monsoon (April 2013), monsoon (August 2014), and post-monsoon (November 2014) seasons were selected to evaluate aspects of temporal variability in greater detail (Figure 6). We divided the seasons into dry (pre-monsoon, post-monsoon) and wet (monsoon) seasons to understand the transport of pollutants via Himalayan valleys in the presence and absence of wet deposition processes. Based on median values, the highest concentrations of both species occurred during the months preceding the monsoon and the lowest were during months of the monsoon. The significantly lower concentrations of both species during the monsoon reflected the combined influences of synoptic easterly airflow that transports a cleaner marine air mass over the region, reduced agricultural residue burning (Sarangi et al., 2014), and more efficient aerosol removal via wet deposition (Dumka et al., 2010). In addition to the possible active wet deposition of $O_3$ precursors during the monsoon season, increased cloudiness during monsoon may also reduce $O_3$ production (Lawrence and Lelieveld, 2010).

The post-monsoon timing of peak BC concentrations observed in previous studies performed in the IGP and Himalayan foothills (see e.g., Tripathi et al, 2005; 2007; Ramchandran et al., 2007; Putero et al, 2015) differs from our observations in the KGV, where we see heightened BC during the pre-monsoon season (Figure 5). These findings are generally in agreement with other high altitude observations. For example, the Nepal Climate Observatory-Pyramid (NCO-P) station at the 5079 m asl in the Himalaya has also shown high seasonal differences for BC and $O_3$ between pre-monsoon (0.444 (±0.443) µg m$^{-3}$ BC; 61 (±9) ppbv $O_3$) and monsoon (0.064 (±0.101) µg m$^{-3}$ BC; 39 (±10) ppbv $O_3$) (Cristofanelli et al., 2010, Marinoni et al., 2013) (Table 1). Our results therefore indicate a lagged peak in BC and $O_3$ within the KGV and presumably other deep Himalayan valleys, as compared to sites within the IGP.

### 3.4 Local winds as drivers of BC and O₃ transport in the KGV

Figure 6 shows the time series of BC and $O_3$ during an individual month in each season (April/pre-monsoon, August/monsoon, November/post-monsoon). For April, BC concentrations peaked at 0700 LST (Figure 6) when wind velocities were low (Figure 6). This peak occurred about an hour later in November, with both periods experiencing a decrease in BC over the rest of the morning as wind speeds increased and diluted local emissions (Figure 4). Thereafter, BC concentrations increased over the afternoon and early night, reaching secondary peaks near midnight LST in April and several hours earlier in November and August (Figures 4 and 6). Distinct morning and afternoon peaks in BC concentration were seen in the post-monsoon season when the up-valley wind speeds were relatively weaker than in pre-monsoon season (Figures 4 and 6). The bimodal diurnal distribution of BC concentration in Jomsom was similar to that observed in Kathmandu (Putero et al. 2015) but unlike a singular late afternoon/evening peak seen at high elevation sites (Bonasoni et al., 2010) during non-monsoonal seasons. This illustrates that deep Himalayan valleys are susceptible to diurnal pollution similar to that of urban areas like Kathmandu. The morning peak in BC was most likely due to local pollutants (from household and morning aircraft traffic) in Jomsom and settlements downwind of Jomsom, while the afternoon peak was

likely primarily associated with long range transport in addition to local pollutants. However, the contribution of long range transport relative to local sources of pollutants is not known. At the same time, $O_3$ exhibited a distinct minimum in the early morning with concentrations increasing towards an early afternoon peak – occurring well before BC's afternoon peak. The $O_3$ minimum in the morning provided further evidence that the morning BC peak originated from local sources, as $O_3$ is only formed downwind of pollution sources. Further, the Jomsom station measuring BC and $O_3$ (JSM_STA) was located more than 100 m above the valley floor – where the village of Jomsom sits. As such, we do not expect that local evening emissions would have reached the station at the cessation of up-valley flows and that evening drainage flows, following the valley floor, would remove these local evening emissions down-valley.

Percentage distributions of up-valley and down-valley BC concentrations, fluxes, and net daily fluxes are depicted in Figure 7 and summarized in Supplementary Table 2. Up-valley and down-valley BC concentrations were quite similar – with down-valley concentrations being slightly higher – but were found to be statistically significantly different for all seasons (Figure 7a). However, because wind velocities were relatively stronger during up-valley daytime flows and the duration of up-valley flows was longer than those of down-valley flow, the corresponding up-valley fluxes of BC during daytime were markedly and significantly greater than down-valley fluxes, across all seasons (Figure 7b). These results suggest an oscillatory movement of polluted air within the valley, where polluted air masses are pushed up-valley during daytime and retreat a shorter distance during nighttime. These differences between up-valley and down-valley fluxes yielded significant net daily up-valley fluxes of BC during all seasons (Figure 7c). Because heating would have driven growth of the boundary layer - thereby enhancing the ventilation and dilution of pollutants during daytime relative to night – we infer that the calculated differences between up-valley and down-valley fluxes correspond to the lower limits for net BC fluxes.

Positive up-valley fluxes were consistent with an "alpine pumping" mechanism in the Himalayan valleys and thereby support the hypothesis that these valleys are important pathways for pollution transport. In addition, we estimated substantial net daily mass transport of BC up the valley during the pre-monsoon season: 1.05 kg day$^{-1}$ (based on the average net daily flux) and 0.72 kg day$^{-1}$ (based on the median net daily flux). While preliminary, these estimates provide the first semi-quantitative constraints on the mass transport of BC from the IGP to the high Himalaya through deep valleys. More generally, these fluxes have important implications for the regional cycling of BC throughout southern Asia.

**3.5 Evidence of regional transport episodes in valley concentration**

Along with the regular diurnal and seasonal variability driven by local winds as described above, we also observed anomalous periods – lasting several days to more than a week – during which BC concentrations were significantly greater than the 90$^{th}$ percentile for corresponding annual averages (Table 2). These extended periods of high BC were evidence of large-scale transport from the IGP to the Himalayan foothills in conjunction with local valley winds (Figure 8). Elevated $O_3$ concentrations did not always accompany these long-duration periods of high BC concentrations above 90$^{th}$ percentile. Different atmospheric lifetimes, chemical reactivity, source location, and sinks of BC and $O_3$ may have contributed to these

differences in BC and O₃ concentrations detected in the valley. Table 2 reports the number of days in which O₃ concentration was above 90th percentile within each episode.

We partitioned these episodes into three characteristic patterns based on the relative variability of BC. Pattern A was characterized as a fluctuating daily maximum in BC with peaks that repeatedly exceeded the 90th percentile but with daily minima below the 90th percentile (Figure. 8[Ia]). Pattern B was characterized by a steady buildup of BC concentration over the period of the regional transport episode, with peak BC concentrations over the 90th percentile but without a daily minimum (Figure 8b). Pattern C exhibited a combination of both Patterns A and B during a single regional episode (Figure 8c). A total of 34 regional episodes were identified from January 2013 through June 2015, 47% of which were categorized as Pattern A, 32% as Pattern B, and 21% as Pattern C. The wind speeds at Jomsom during these transport episodes exhibited diurnal variability similar to those during other periods (Figure 8[II]). During the regional transport period in November 2014 (Pattern A), average daily BC concentration was 1.3 µg m⁻³ which is over the 75th percentile (0.9 µg m⁻³) of the BC concentration for the measurement duration. The maximum daily concentration during the period was 3.1 µg m⁻³. However, the corresponding average O₃ was only 28.1ppbv, slightly below the average (29.5 ppbv) for the entire data set (Figure 8[Ia]). The mean BC concentration during a May 2014 Pattern B transport episode was 1.8 µg m⁻³ (Figure 8[Ib]). It was above the 90th percentile (1.5 µg m⁻³) for the entire measurement period while O₃ concentrations were at 49.7ppbv, slightly below the 90th percentile (52.9 ppbv). One of the Pattern C- type transport episodes was identified in May 2013, when the average concentration was well above the 90th percentile for both BC (2.1 µg m⁻³) and O₃ (57.5 ppbv) (Figure 8[Ic]). The diurnal wind pattern in the KGV was conserved during the Pattern A example, but a longer period of up-valley flows occurred during the examples for Patterns B and C (Figure 8).

## 4. Conclusion

This study provides new in-situ observational evidence of the role of a major Himalayan valley as an important pathway for transporting air pollutants from the IGP to the higher Himalaya. We found that concentrations of BC and O₃ in the KGV exhibited systematic diurnal and seasonal variability. The diurnal pattern of BC concentrations during the pre- and post-monsoon seasons were modulated by the pulsed nature of up-valley and down-valley flows. Seasonally, pre-monsoon BC concentrations were higher than in post-monsoon season. We also found that morning and afternoon peaks in the post-monsoon season were more pronounced than those of pre-monsoon season, likely due to the relatively lower wind speeds during post-monsoon. Significant positive up-valley fluxes of BC were measured during all seasons, and preliminary flux estimates (which require a more robust estimate in future work) show the efficiency and magnitude of pollutant transport up the valley. During episodes of regional pollution over the IGP, relatively higher concentrations of BC and O₃ were also measured in the KGV.

The frequency and magnitude of pollution events highlighted in the paper need to be studied for a longer period in order to understand the associated interannual variability. In addition, future work should focus on understanding the vertical and

horizontal distribution of particulate matter and ozone in the Himalayan region, and their impacts on the radiative budget, the
320 ASM, and regional climate.

**Acknowledgments.**

We would like to acknowledge our field assistant in Nepal, Buddhi Lamichhane, who helped us in various stages of the study, as well as the logistic and administrative support and internet at the Jomsom station provided by Nepal Wireless. Financial support was provided by the National Aeronautics and Space Administration NNX12AC60G, and additional field
support was provided by ICIMOD's Atmosphere Initiative. The authors are very thankful for comments from William Keene, Jennie Moody, and Kyle Davis.

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

**Tables and Figures**

Table 1. Comparison of BC and $O_3$ concentrations (mean ±standard deviation) between high elevation Himalayan site NCO-P CNR and JSM_1.

| Sites | Altitude (m) | Co-ordinates | Season | BC ($\mu g\ m^{-3}$) | $O_3$ (ppb) |
|---|---|---|---|---|---|
| NCO-P | 5079 | 27.95°N, 86.81° E | Pre-monsoon | 0.32 (±0.34) | 60.9 (±8.4) |
| (Bonasoni et al., 2010) | | | Monsoon | 0.05 (±0.06) | 38.9 (±9.0) |
| | | | Post-monsoon | 0.14 (±0.08) | 46.3 (±5.0) |
| JSM_1 | 2800 | 28.87°N, 83.73° E | Pre-monsoon | 0.90 (±0.45) | 39.5 (±8.23) |
| | | | Monsoon | 0.21 (±0.24) | 25.1 (±6.48) |
| | | | Post-monsoon | 0.71 (±0.42) | 31.4 (±4.50) |

**Table 2. List of enhanced BC episodes observed at JSM_STA and the concurring regional sources from MODIS (\* Data is from January-July; $^+$ Incomplete $O_3$ data).**

| 2013 ($90^{th}$ percentile : BC = 1.53 µg m$^{-3}$ and $O_3$ = 49.5 ppbv) | | | |
|---|---|---|---|
| **Month** | **Episode Length** | **Episode type** | **Days with $O_3$ mixing ratio above $90^{th}$ percentile** |
| Jan | $6^{th} - 15^{th}$ | C | 1 |
| | $28^{th} - 31^{st}$ | C | $1^+$ |
| Feb | Jan $28^{th}$ – Feb $2^{nd}$ | C | $1^+$ |
| Mar | $1^{st} - 3^{rd}$ | A | 0 |
| | $6^{th} - 9^{th}$ | A | 0 |
| | $11^{th} - 13^{th}$ | B | 0 |
| | $18^{th} - 27^{th}$ | B | 4 |
| Apr | $6^{th} - 10^{th}$ | B | 5 |
| | $12^{th} - 14^{th}$ | B | 3 |
| | $27^{th} - 30^{th}$ | A | 4 |
| May | $3^{rd} - 11^{th}$ | B | 9 |
| Oct | $28^{th} - 30^{th}$ | A | No data |
| Nov | $1^{st} - 5^{th}$ | A | No data |
| | $23^{rd} - 30^{th}$ | A | $0^+$ |
| Dec | $17^{th} - 24^{th}$ | C | $0^+$ |
| 2014 ($90^{th}$ percentile: BC = 1.60 µg m$^{-3}$ and $O_3$ = 56.8 ppbv) | | | |
| **Month** | **Episode Length** | **Episode type** | **Days with $O_3$ mixing ratio above $90^{th}$ percentile** |
| Jan | $13^{th} - 15^{th}$ | A | No data |
| Feb | Jan $31^{st}$ – Feb $1^{st}$ | A | No data |
| Mar | $14^{th} - 16^{th}$ | A | No data |
| Apr | $5^{th} - 8^{th}$ | B | 2 |
| | $10^{th} - 30^{th}$ | C | $15^+$ |
| May | Apr $10^{th}$ – May $1^{st}$ | C | $15^+$ |
| | $8^{th} - 13^{th}$ | B | 6 |
| | $17^{th} - 23^{rd}$ | C | $5^+$ |
| Jun | $5^{th} - 8^{th}$ | B | 3 |
| | $10^{th} - 17^{th}$ | B | 8 |
| Nov | $15^{th} - 30^{th}$ | A | 0 |
| Dec | Nov $15^{th}$ – Dec $8^{th}$ | A | 0 |
| 2015 ($90^{th}$ percentile: BC = 1.47 µg m$^{-3}$ and $O_3$ = 49.7 ppbv) | | | |
| **Month** | **Episode Length** | **Episode type** | **Days with $O_3$ mixing ratio above $90^{th}$ percentile** |
| Jan | $15^{th} - 17^{th}$ | A | No data |
| | $21^{st} - 25^{th}$ | A | No data |
| Feb | $21^{st} - 26^{th}$ | A | No data |
| May | $6^{th} - 9^{th}$ | A | $0^+$ |
| | $18^{th} - 26^{th}$ | A | 8 |
| | $29^{th} - 31^{st}$ | B | 3 |
| Jun | $6^{th} - 12^{th}$ | B | 7 |

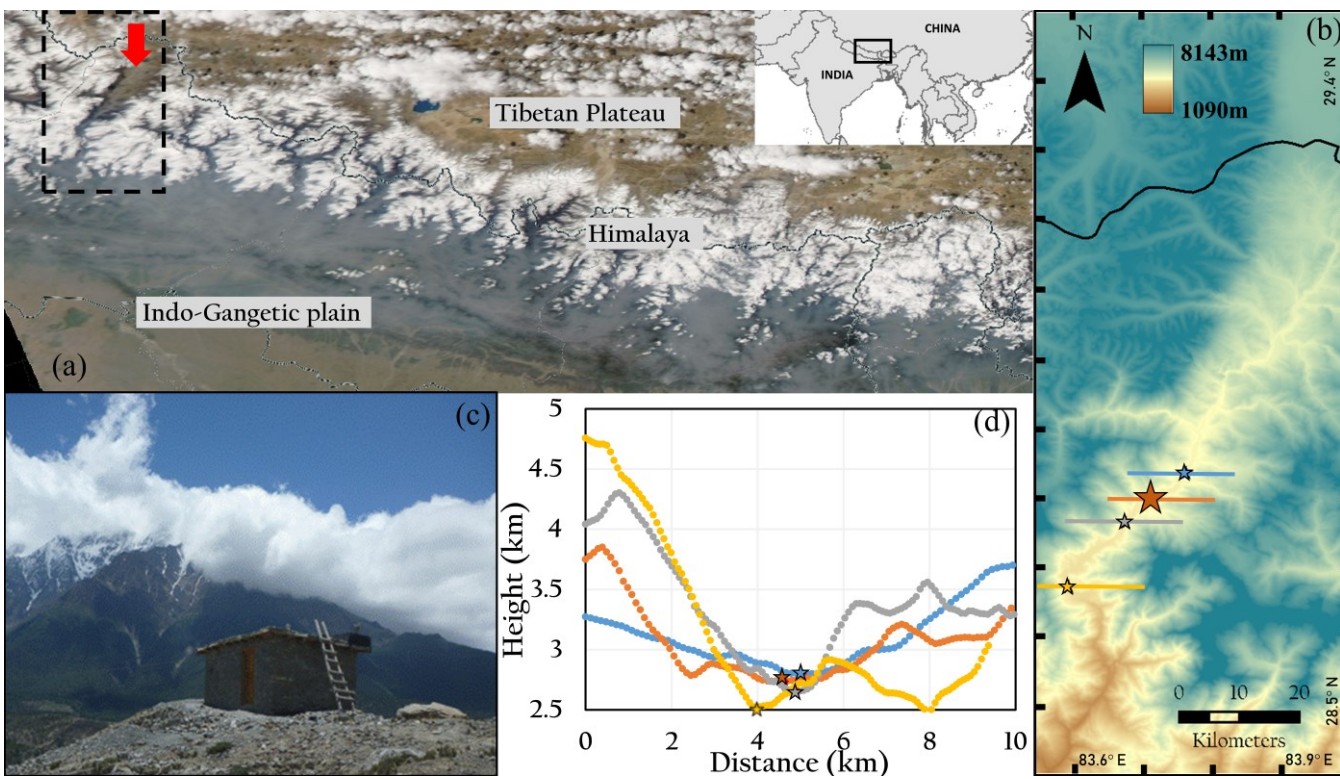

**Figure 1. (a) NASA Worldview image from November 4$^{th}$ , 2014 depicting thick haze intruding the Himalayan foothills with red arrow over the KGV. Inset map in the top-right corner of panel (a) shows the location of the Worldview image. Dashed box shows the location of panel (b). (b) Expanded scale of the KGV showing station locations of Lete (LET_AWS) near the entrance of the valley (yellow star); Marpha (MPH-AWS) in the core region (gray star); the Jomsom (JSM_STA) sampling station for BC and O$_3$ and the two associated AWS sites (JSM_1 and JSM_2) in the core region (orange star); and Eklobhatti (EKL_AWS) near the exit (blue star). (c) The atmospheric observatory at Jomsom (JSM_STA) (28.87° N, 83.73° E, 2900 m asl). (d) Valley cross-sectional elevation profile at the station locations.**

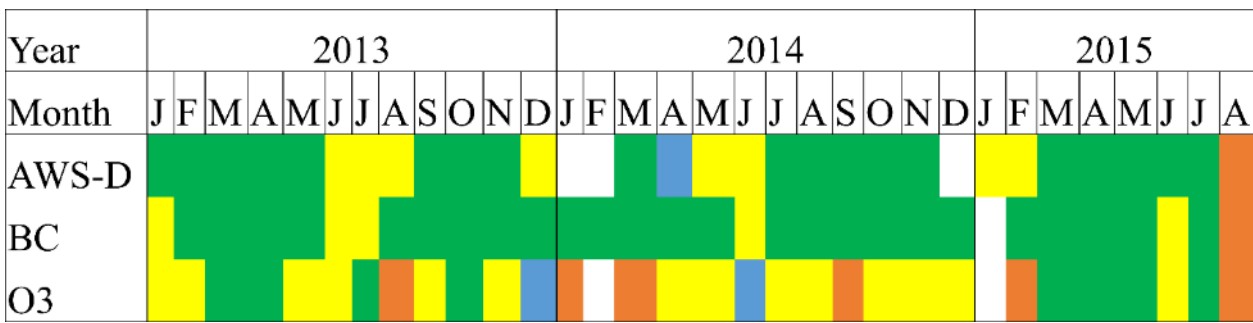

**Figure 2. Data timeline for JSM_2 AWS, BC, and O₃ measurements in Jomsom. Green indicates complete data; blue indicates only a few missing data points; yellow indicates more than 15 days of data; orange indicates less than 15 days of data, and white indicates no data.**

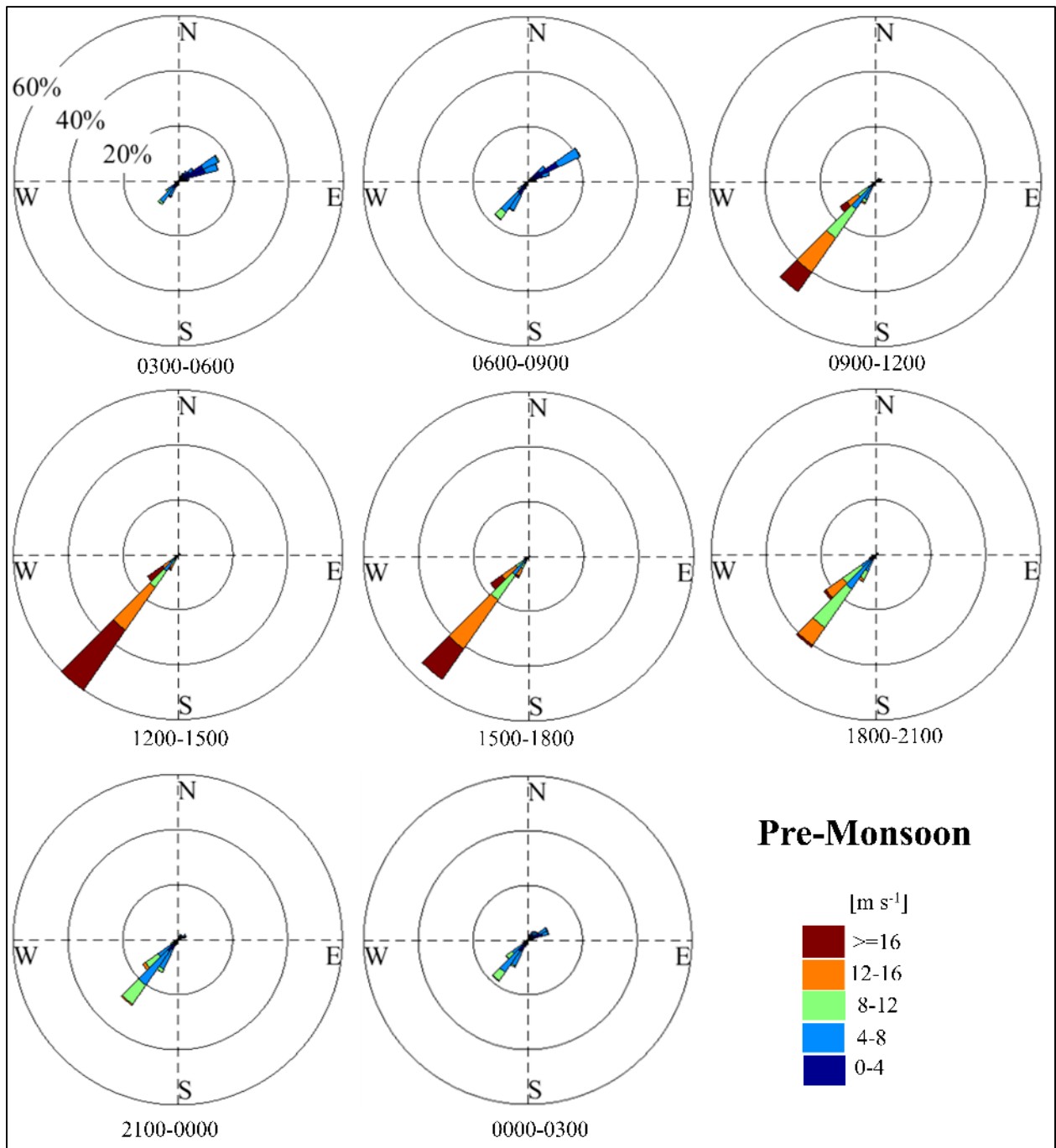

**Figure 3. Wind rose for pre-monsoon season – binned into 3-hour increments depicting diurnal evolution in wind speed and direction at JSM_2.**

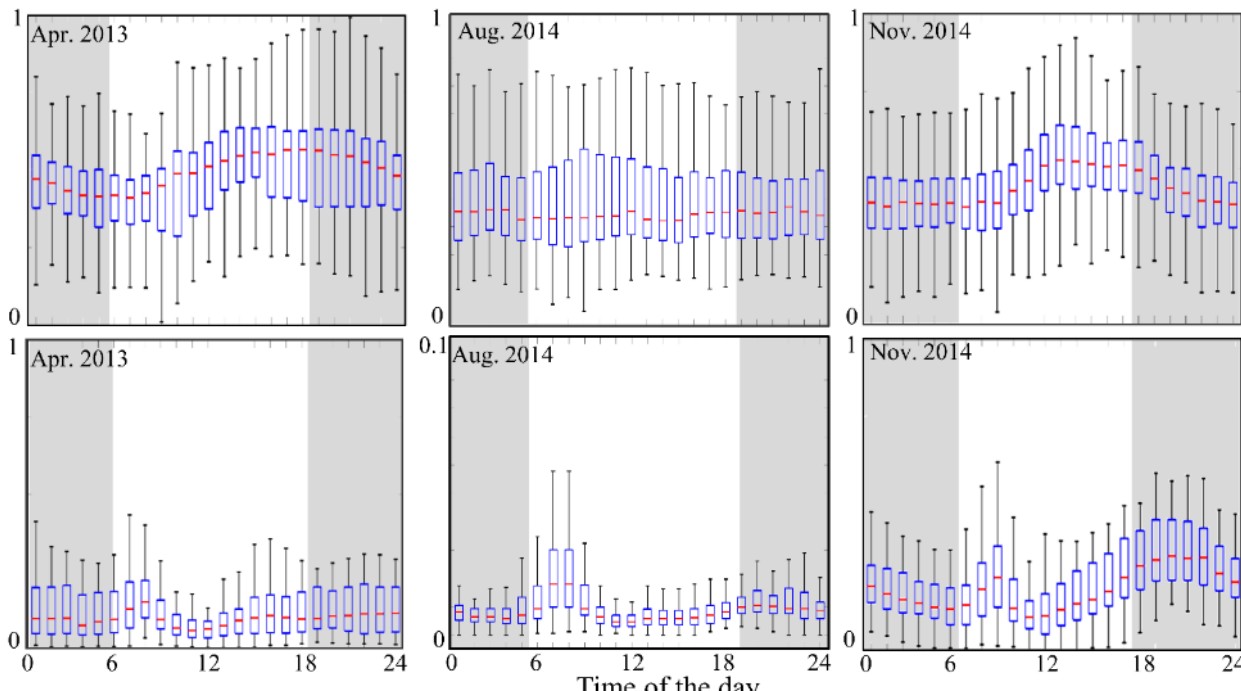

**Figure 4. Box and whisker plots depicting the 90th, 75th, 50th, 25th, and 10th percentiles for normalized diel variability in O₃ (upper panels) and BC (lower panels) at JSM_STA during April 2013 (pre-monsoon), August 2014 (monsoon), and November 2014 (post-monsoon). Scale for August 2014 is from 0 to 0.1.**

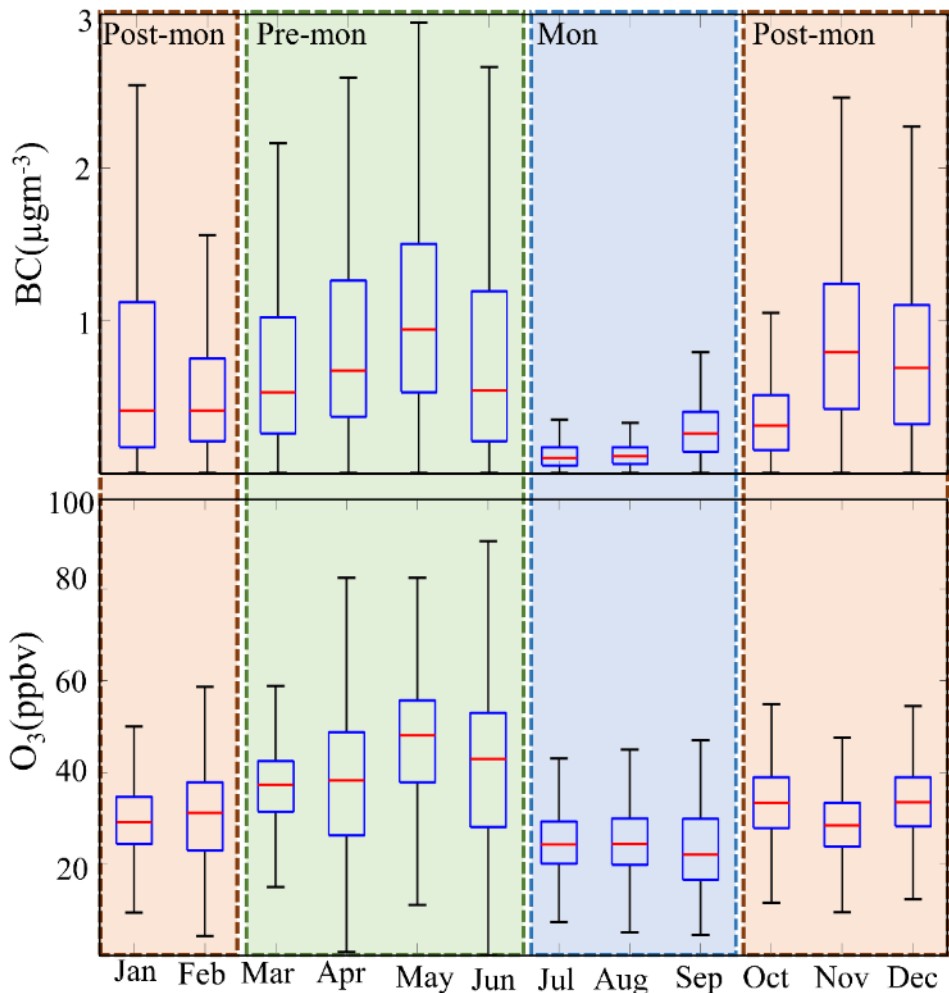

**Figure 5. Box and whisker plots depicting the 90th, 75th, 50th, 25th, and 10th percentiles for monthly concentrations of BC (upper panel) and O₃ (lower panel) between January 2013 and August 2015 at JSM_STA. Orange shaded areas indicate the post-monsoon season (Post-mon), green shaded area indicates the pre-monsoon (Pre-mon) and blue are indicates the monsoon (Mon) season.**

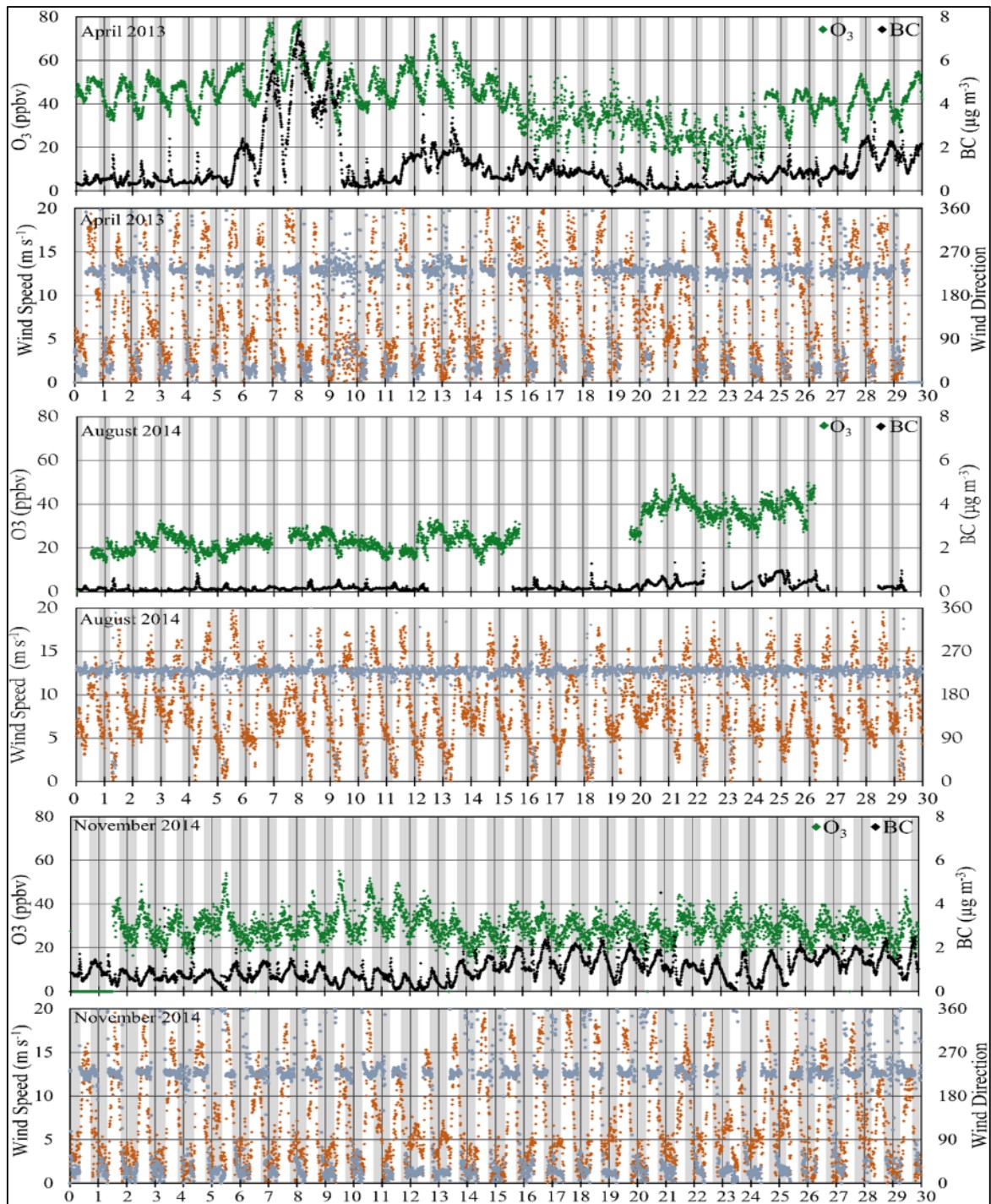

**Figure 6. Variation in O₃, BC, and associated wind speed and direction at JSM_STA during (a) April 2013 (pre-monsoon), (b) July 2015 (monsoon), and (c) November 2014 (post monsoon). The grey shaded area denotes night.**

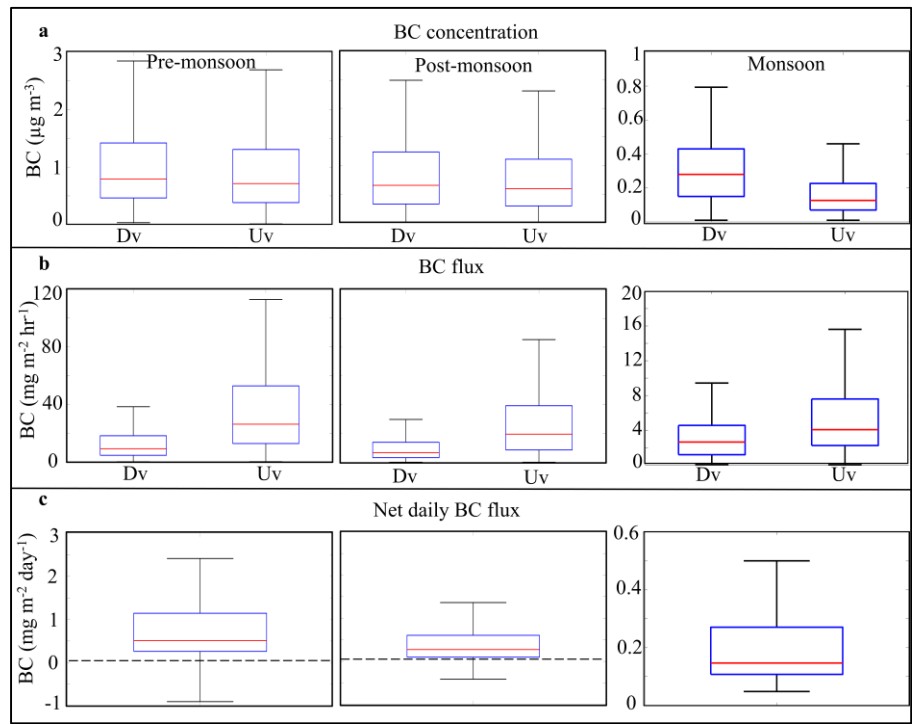

**Figure 7. (a) BC concentration distribution with down-valley (Dv) and up-valley (Uv) flows in Jomsom, (b) calculated Dv and Uv flux for each season, (c) Net daily flux per season. The dotted line is panel c marks 0 mg m⁻² day⁻¹. The red line represents 50th percentile, the edge of the box 25th and 75th percentile while the whiskers show maximum and minimum values. Scale for monsoon season is 0 to 1 μg m⁻³ for BC concentrations, 0-20 mg m⁻² hr⁻¹ for BC flux and 0 to 0.6 mg m⁻² day⁻¹ for net daily BC flux.**

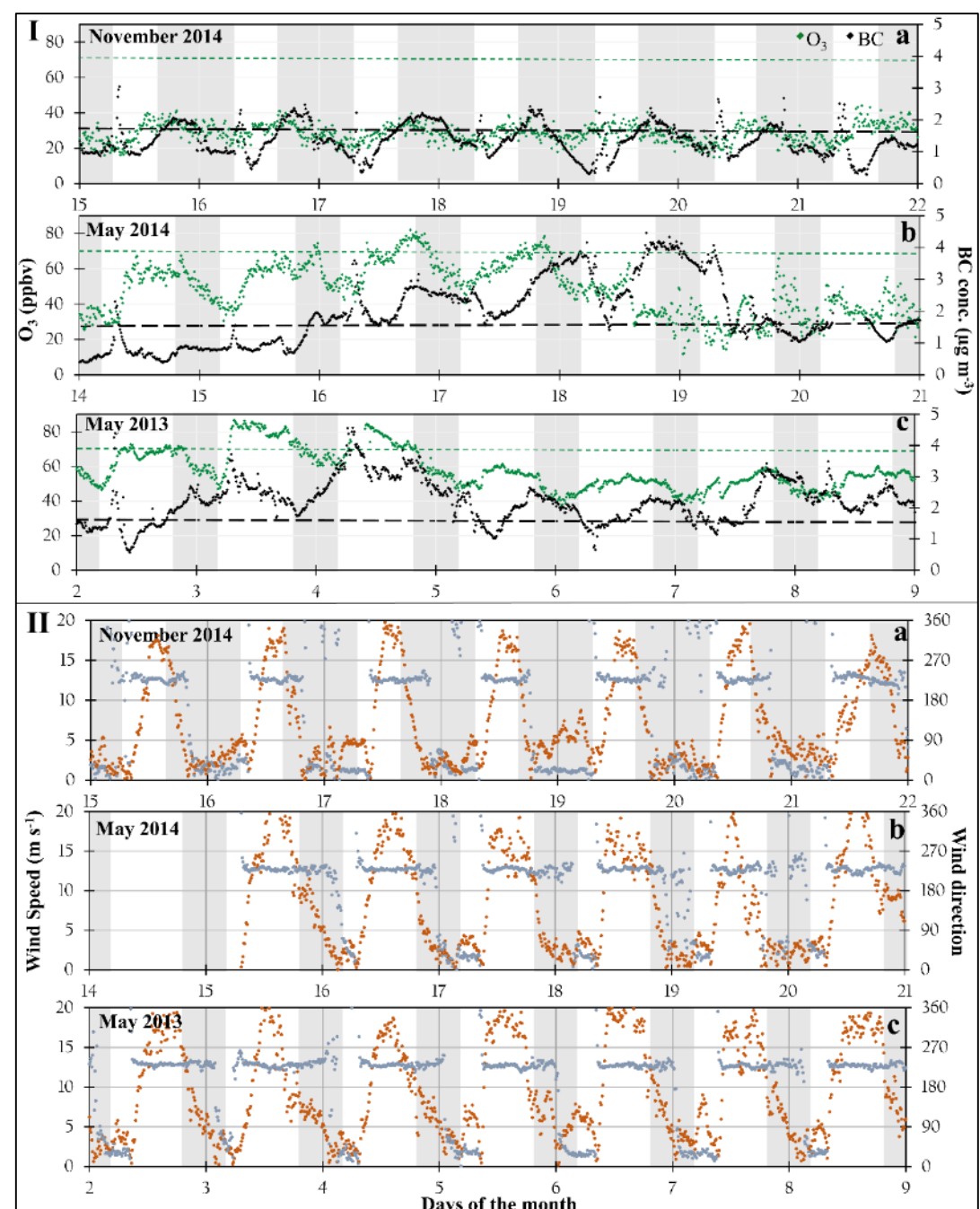

Figure 8. (I) Examples of extended periods with relatively high BC and $O_3$ concentrations at JSM_STA during November 2014 (Pattern A) and May 2014 (Pattern B) and May 2013 (Pattern C). The dashed black and green lines depict two-year averages for BC and $O_3$, respectively. (II) Corresponding wind direction and wind speed during the high BC and $O_3$ episodes. The orange dots represent wind speed, and the blue dots represent wind direction.