# Peer review of "Transport of regional pollutants through a remote trans-Himalayan valley in Nepal"

_Atmospheric Chemistry and Physics, 2016_

## Referee Comment (RC1) · Anonymous Referee #1 · 6 Dec 2016

GENERAL COMMENTS

This paper by Dhungel et al., 2016, provides a first characterization of the variability of ozone and equivalent BC at a measurement site located in a Himalayan valley. Until now only sparse continuous measurements are available in the Himalayas region. Thus the data presented in this work can be considered of high interest for the advancement of knowledge about SLCF (short-lived climate forcers) variability in the Himalaya and about the emissions and atmospheric processes able to affect them. Unfortunately, it is not clear from the paper, which data coverage is available over the whole (2.5-year long) investigation period.

However, the paper suffers of major deficiencies that prevent publication in this current form. Indeed, the paper only provides a basic characterization of typical seasonal and

diurnal variability of O3 and BC: only a tentative attribution of the observed variability in terms of valley wind regime. No information about the role of synoptic-scale transport variability is provided. The data analysis is basic and lacking of statistical analysis. The possible impact of open fire emissions in the IGP and Himalaya foothills should be better assessed by carrying out a systematic analysis. At least, the three case studies presented in Figure 6 should be better explored (as an instance by investigating them by using air-mass transport modeling and a better use of satellite data) and extended (e.g. no information is provided about the frequency by which the three "regimes" were observed over the whole measurement period). The occurrence of open fires is a typical feature of the pre-monsoon season in the Himalaya foothills. Why the transport of fire emission along the valley is observed only in a few cases? Which are the factors triggering the transport of open fire emissions?

Some previous works already extensively investigated the role of thermal wind circulation and open fire emission in affecting atmospheric composition in Himalaya (e.g. Bonasoni et al., ACP, 2010; Dumka et al., ACP, 2015; Lüthi et al., ACP, 2015; Xu et al., ACP, 2015; Raatikainen et al., Atmos. Env. 2014; Hyvärinen, ACP, 2011a,b). It should be great if the authors can discuss their results as a function of these previous investigations even clarifying the scientific advance of their study in respect to these previous works. As an instance, the diurnal behaviors of BC and O3 are significantly different from those observed at other Himalayan site (e.g. NCO-P, or Naintal ,see Bonasoni et al., 2010; Dumka et al., ACP, 2015) which reports eqBC increase from early morning and peaking in the afternoon. The authors should better motivate these differences. Finally, as also admitted by the authors, a not negligible influence on the observed behaviors could relate to local emissions (see the BC peak observed in the morning). It should be important (and interesting) that this local contribution is isolated and quantified before discussing eqBc and O3 variability.

Moreover, I cannot able to find along the paper a real proof about the transport of pollution from IGP to TP: the paper only presents observations inside the Himalayas

valley, thus the export of this pollution to TP is just a speculation at this stage...

Finally, I strongly suggest a language revision by a native-speaking English person.

SPECIFIC COMMENTS

Pag 3, line 101. Extensive investigation of the role of valley wind system in favoring the transport of SLCFs to Himalayas was presented by Bonasoni et al., 2010 and reference therein. These researches can be profitably cited at this point of Introduction other than reporting the (rather dated) works from Alpine region. Also this work can be profitably cited: Quantification of topographic venting of boundary layer air to the free troposphere. S. Henne, M. Furger, S. Nyeki, M. Steinbacher, B. Neininger, S. F. J. de Wekker, J. Dommen, N. Spichtinger, A. Stohl, and A. S. H. Prévôt. Atmos. Chem. Phys., 4, 497-509, doi:10.5194/acp-4-497-2004, 2004.

Pag. 4, line 133: actually, the intrusion of the haze is not so visible from Figure 1a.

Pag. 4, line 138: the description of the valley orientation is difficult to follow. Some of the described features (e.g. Eastward orientation at Jomstom) cannot be captured by Figure 1. I would suggest to add to Figure 1 a more detailed map of the measurement site.

Pag 5, line 148: please add to Table 1 a column with measured parameters

Pag 5, line 151: actually "equivalent BC" is measured by MAAP.

Pag 5, line 158: please substitute "attenuation" by "absorption". For O3 and eqBC, please provide indication about measurement uncertainty and QA/QC procedures.

Pag 5, line 161: please indicate the percentage of data available over the period January 2013 – August 2015. Please, remove the sentence "Measurements of carbon monoxide..." (no CO data were presented/discussed in the paper).

Pag 5, line 164: no winter season has been identified?

Pag 5, line 169: I would skip "about 10 meters above and"

Pag 5, line 175: I cannot understand this kind of normalization. Why did you not report actual eqBc and O3 values? You should simply report the averaged seasonal diurnal variation of O3 and eqBC obtained by subtracting averaged monthly values from hourly values.

Figure 2: I would like to see the percentiles for each single month. This would provide also information about year-to-year variability.

Pag 6, line 189: please provide references. Possible reduced domestic emissions related to less domestic heating?

Pag 6, line 197: "Seasonal variability....broad regional pattern". I do not agree. In the IGP, BC is maximized during winter months (December- January), while in Himalayas (and also at your station) the values are higher during pre-monsoon! (see also Ratikainen et al., 2014 Atmos Env).

Pag 6, line 209: "These differences in...". Not clear: what differences?

Pag 6, line 213: The works by Ratikainen et al., 2014 AtmosEnv can be cited here

Section 3.2: this discussion is mainly qualitative. No statistical analysis have been applied and it is difficult to discern if the observed features are statistically significant. I suggest to add a line describing the mean average values with statistical confidence level (this would help in understand if the observed peak and minima are robust features). I would add to these plots the analogous for wind direction and speed to clearly correlate wind regime with O3 and eqBc variability. In any case, the results are based just on the analysis of 3 single months of observations. A comment for taking into account the possible intra-seasonal and year-to-year variability should be added. Your measurement period is 2.5 year-long. Why you did not use all the available data?

Pag 7, line 219: "peaked in the early afternoon". I would say "at noon"! This can be an hint for local photochemical production...

[Figure]

Pag 7, line 220. "Finally,...0 to 1 (Fig. 3)", I cannot be able to understand this sentence...Maybe you would suggest that diurnal variability account for the most part of the overall O3 variability? In the case, this is a point that should be better stressed. Can you quantify it?

Pag 7, line 223: "increased rapidly following sunrise"...because later in the manuscript, you suggested that this peak can be related to local emissions, I would change with "increased rapidly in the early morning".

Pag 7, line 225-229: Again, this sentence is not clear to me. See the same comment for ozone.

Pag 7, line 230: you should also mention air-mass transport. At a remote site, if local emissions are really negligible (I'm not totally convinced about this for your site, see your following sentence about eqBC), I would expect that the contribution by transport is the most important one!

Pag 7, line 236: the secondary peak (in the evening from 19 to 21)is visible only during the post-monsoon. Please comment. Does this peak be related with domestic emissions (e.g. domestic cooking or heating)?

Pag 7, line 239: "Up-valley...Alpine pumping". As mentioned before, many works in Himalayas investigated the role of valleys as channel of anthropogenic pollution. Please consider them and comment your results as a function of these previous works.

Section 3.3: the expected outcome from this Section is not clear. Why did you show just 6 days of data at JSM_2, when more than two years of meteorological data are available at the "core" site where also O3 and eqBC data were available? You must show these data! Moreover, if I'm not wrong, JSM_2 is located 1000 m above the "core" site. Thus, which is the goal of showing these data?

Section 3.4: Legend is missed in Figure 6. I suspect that blue dots represent O3 but you have to add a legend! Basically, this section repeat the same concept about diurnal

variability already reported by Section 3.2. . .

Pag 8, line 271: "This peak occurred about an (one) hour later during the post-monsoon period". Looking at Figure 3, this seems not true! eqBc peak at 8:00 AM during all the seasons. However, it is important to evaluate the robustness and origin of this peak. Looking at the eqBc time series reported in Figure 6, it looks that the early morning peak is related to "spiky" observations, very likely related to local emissions. This is particularly evident during post-monsoon, when these "spikes" where observed during the diurnal minima of eqBC. This feature can be of a certain interest to evaluate the local emissions to the "pristine" Himalayan environment, but I would neglect it for the analysis of transport processes affecting O3 and eqBC variability.

Pag 8, line 275: ". . ..decreasing concentration with increasing wind speeds are consistent with expectation based on dilution". I think that the decrease on eqBC observed during midday can be associated not only to dilution in a more developed PBL of local emissions, but also to the fact that air-masses reaching the site from the lower valley are still not enriched in pollutant. . .Indeed, you observed eqBC (and O3) increase in the afternoon/evening (when both PBL height decrease and air-masses richer in pollution could reach the site from longer distance). You should roughly evaluate the distance of eqBC emissions by analysing wind speed at the measurement sites. . .

Pag 8, line 285. This detailed description of local wind regime needs a more detailed map on Figure 1!

Pag 9, line 294: "Nocturnal decupling of the boundary layer preserves the concentration of. . .". Not clear: what do you mean with "decoupling PBL"? Decoupling from what? Section 3.4: I assume that this Section should be "Evidence of LONG-RANGE transport episodes. . .". In this section you discuss three typical regimes of O3 and eqBC variability. However, I would like to see a more detailed description of the main features of each single regime (basically how are you able to distinguish among them?) and a systematic assessment of their occurrence and impact on eqBC and O3 variability.

E.g. which is the frequency of occurrences of these regimes on a seasonal basis? Are you able to objectively identify (by some selection criteria) the occurrence of that regimes? Can you able to compare your results with previous studies? (e.g. Putero et al., Environ. Poll. 2013)

Pag 9, line 304: actually you did not show any evidence of transport to TP! I think this is just a (reasonable) speculation. Maybe you can discuss this possibility in the conclusions Section.

Pag 9, line 310: I think that the relationship between eqBc , O3 and fire emissions in the IGB is only qualitative and deserve more analyses. At least a correlation between the temporal variation of the fire number in the IGP and the eqBc and O3 at the measurement site must be showed. Moreover, air-mass transport analysis (i.e. back-trajectories or dispersion plume) corroborating the transport towards the measurement site region should be showed (the same is valid for case B). For case B and C, eqBc is maximized during night-time, when down-valley winds are expected in a mountain valley and cleaner air-masses from upper layers should be present (add the behaviors of wd and ws!). Please, comment. Moreover, the different co-variability (correlation) between O3 and eqBC should be better investigated and commented. During case B, the diurnal variability of O3 and eqBC appeared to be minimized. Can this indicate the role of far (fires?) emissions instead of regional/local ones? The increase of daytime minima value of eqBc and O3, would indicate a build-up of pollution. Is this build-up only limited to the valley or did it extend to the foothills? Maybe time series of satellite MODIS data can help. . .

Section 3.4.2 is confused and not provides important information: thus, it can be skipped. STE discussion at this point is a little bit out of the scope of the paper. A rigorous assessment of STE contribution to eqBC and O3 variability would deserve a specific investigation. Moreover, I would note that the focus on the monsoon season is of limited interest, since it is well assessed that summer monsoon is the season during which transport of stratospheric air-masses to the lower troposphere is minimized over

the Himalayan region (see e.g. Putero et al., 2016, Ohja et al., ACPD, 2016).

The conclusions are not robust. Actually you did not demonstrate the transport of pollutant from IGP to TP but just the transport of pollution up to the valley. Your main results (seasonal and diurnal O3 and eqBC variability, influence of open fires)should be better commented in the framework of the most recent studies on the topic.

---

## Referee Comment (RC2) · Anonymous Referee #2 · 10 Dec 2016

The manuscript reports data that is collected recently over a high altitude site and does not bring in any new insights or results. As the authors write, many results from the same group have been published recently. What is the relevance of connecting ozone and BC is not clear.

Many aspects are very loosely dealt with and mentioned in passing. References are missing.

Lines 285 - how this phenomenon can occur? What about lifetime of BC aerosols?

Conclusion - with the limited scope of the study the conclusions are far fetched.

---

## Author Response (AR1)

We thank the two anonymous reviewers for their helpful comments on and recommendations for improvement of the subject manuscript. We have made major edits to the submitted version, which has helped improve the clarity of our presentation. The responses to Referee #1 and #2 are included in this document. Each comment is listed below followed by our response to that comment (in italics).

**Referee #1**

GENERAL COMMENTS

This paper by Dhungel et al., 2016, provides a first characterization of the variability of ozone and equivalent BC at a measurement site located in a Himalayan valley. Until now only sparse continuous measurements are available in the Himalayas region. Thus the data presented in this work can be considered of high interest for the advancement of knowledge about SLCF (short-lived climate forcers) variability in the Himalaya and about the emissions and atmospheric processes able to affect them. Unfortunately, it is not clear from the paper, which data coverage is available over the whole (2.5-year long) investigation period.

However, the paper suffers of major deficiencies that prevent publication in this current form. Indeed, the paper only provides a basic characterization of typical seasonal and diurnal variability of O3 and BC: only a tentative attribution of the observed variability in terms of valley wind regime. No information about the role of synoptic-scale transport variability is provided. The data analysis is basic and lacking of statistical analysis. The possible impact of open fire emissions in the IGP and Himalaya foothills should be better assessed by carrying out a systematic analysis. At least, the three case studies presented in Figure 6 should be better explored (as an instance by investigating them by using air-mass transport modeling and a better use of satellite data) and extended (e.g. no information is provided about the frequency by which the three "regimes" were observed over the whole measurement period). The occurrence of open fires is a typical feature of the pre-monsoon season in the Himalaya foothills. Why the transport of fire emission along the valley is observed only in a few cases? Which are the factors triggering the transport of open fire emissions?

Some previous works already extensively investigated the role of thermal wind circulation and open fire emission in affecting atmospheric composition in Himalaya (e.g. Bonasoni et al., ACP, 2010; Dumka et al., ACP, 2015; Lüthi et al., ACP, 2015; Xu et al., ACP, 2015; Raatikainen et al., Atmos. Env. 2014; Hyvärinen, ACP, 2011a,b). It should be great if the authors can discuss

their results as a function of these previous investigations even clarifying the scientific advance of their study in respect to these previous works. As an instance, the diurnal behaviors of BC and O3 are significantly different from those observed at other Himalayan site (e.g. NCO-P, or Naintal ,see Bonasoni et al., 2010; Dumka et al., ACP, 2015) which reports eqBC increase from early morning and peaking in the afternoon. The authors should better motivate these differences. Finally, as also admitted by the authors, a not negligible influence on the observed behaviors could relate to local emissions (see the BC peak observed in the morning). It should be important (and interesting) that this local contribution is isolated and quantified before discussing eqBc and O3 variability.

Moreover, I cannot able to find along the paper a real proof about the transport of pollution from IGP to TP: the paper only presents observations inside the Himalaya valley, thus the export of this pollution to TP is just a speculation at this stage. . .

Finally, I strongly suggest a language revision by a native-speaking English person.

> *In response to the reviewer's comment, we have: 1) added additional references to relevant previously published work; 2) added qualifications regarding interpretation of pollutant transport from the IGP; 3) statistically evaluated differences between the daytime and nighttime concentrations across seasons; 4) added seasonal flux patterns and calculated the net daily fluxes between different seasons with statistical analysis; 5)clarified characterization of the different types of transport episodes 6)added HYSPLIT back trajectories to the examples of transport episode.*

SPECIFIC COMMENTS

Page 3, line 101. Extensive investigation of the role of valley wind system in favoring the transport of SLCFs to Himalayas was presented by Bonasoni et al., 2010 and reference therein. These researches can be profitably cited at this point of Introduction other than reporting the (rather dated) works from Alpine region. Also this work can be profitably cited: Quantification of topographic venting of boundary layer air to the free troposphere. S. Henne, M. Furger, S. Nyeki, M. Steinbacher, B. Neininger, S. F. J. de Wekker, J. Dommen, N. Spichtinger, A. Stohl, and A. S. H. Prévôt. Atmos. Chem. Phys., 4, 497-509, doi:10.5194/acp-4-497-2004, 2004.

> *The references section has been revised as follows:*

> *"In the European Alps, prevailing wind systems in the mountain river valleys funnel polluted air from peripheral source regions to high elevations in a phenomenon known as "Alpine Pumping" (Weissmann et al., 2005). Under fair weather conditions during daytime, the upslope winds are capable of transporting significant pollutants and moisture into the free troposphere (Henne et al., 2004). Relative to air over plains, the*

*air within the valleys heats and cools more quickly (Steinacker, 1984). The resultant differences in temperature create gradients in pressure and density, which in turn drive transport of air from the plains to higher-elevations during the daytime (Reiter and Tang 1984, Whiteman and Bian, 1998; Egger et al., 2000). Numerous studies have looked at the possibility of the transport of pollutants from the IGP to the Himalayan foothills (Pant et al., 2006; Dumka et al., 2008; Komppula et al., 2009; Hyvärinen et al., 2009; Ram et al., 2010; Brun et al., 2011; Gautam et al., 2011; Srivastava et al., 2012). Other studies show that pollutants have the potential to reach not only the foothills of the Himalaya but also higher elevations (Bonasoni et al., 2010; Decesari et al., 2010; Marinoni et al., 2010). In addition, studies have shown the obstruction of flow caused by the high Himalaya which intensifies the effect of pollution over the IGP that are visible in satellite imagery especially during pre-monsoon seasons (Singh et al., 2004; Dey and Di Girolamo, 2010; Gautam et al., 2011). Though there is evidence of existence of similar source pollutants, both regional and local, in the foothills and the higher altitude sites (Raatikainen et al., 2017; Raatikainen et al., 2014; Srivastava et al., 2012,), observational evidence of mechanisms and pathways facilitating such transport via Himalayan valleys is lacking. Our data fills this gap by characterizing the role of the wind system within a deep Himalayan valley in transporting pollutants from the IGP to the high mountains."*

Page 4, line 133: actually, the intrusion of the haze is not so visible from Figure 1a.

*The image in figure 1a has been zoomed in to better show the intrusion of haze.*

Page 4, line 138: the description of the valley orientation is difficult to follow. Some of the described features (e.g. Eastward orientation at Jomstom) cannot be captured by Figure 1. I would suggest to add to Figure 1 a more detailed map of the measurement site.

*We have replaced table 1 with a new figure 2, which includes a clear map of the valley.*

Page 5, line 148: please add to Table 1 a column with measured parameters

*The measured parameters are now included in the new figure 2.*

Page 5, line 151: actually "equivalent BC" is measured by MAAP.

*The text now reads:*

*"Equivalent black carbon (hereafter referred to as BC)…" is specified when first used and, for efficiency, the acronym BC is used thereafter.*

Page 5, line 158: please substitute "attenuation" by "absorption". For O3 and eqBC, please provide indication about measurement uncertainty and QA/QC procedures.

We have *changed attenuation to absorption. Additional description on QA/QC procedures have been added.*

Page 5, line 161: please indicate the percentage of data available over the period January 2013 – August 2015. Please, remove the sentence "Measurements of carbon monoxide. . ." (no CO data were presented/discussed in the paper).

*A data timeline has been added as supplementary table 1 and we have removed CO measurements from the sentence.*

Page 5, line 164: no winter season has been identified?

*Our analysis focused on variability in BC and O3 variability during wet (monsoon) and dry (pre-monsoon and post-monsoon) seasons. Consequently, identification and interpretation of variability during the winter season was not directly relevant.*

Page 5, line 169: I would skip "about 10 meters above and"

*We prefer to retain the clarification that meteorological parameters were measured 10 meters above ground to ensure that readers know that AWS were above the surface layer.*

Page 5, line 175: I cannot understand this kind of normalization. Why did you not report actual eqBc and O3 values? You should simply report the averaged seasonal diurnal variation of O3 and eqBC obtained by subtracting averaged monthly values from hourly values.

*The approach we employ here to characterize diurnal variability normalizes for day-to-day variability in concentrations, quantitatively captures the frequency distributions in normalized cycles, and is widely used in the literature (e.g., Sander et al., 2003, ACP; Fischer et al., 2006; JGR, Keene et al., 2007, JGR; Smith et al., 2007, JGR, Young et al., 2013, JGR; among others). Cycles derived from average values typically dampen the relative range of actual diurnal variability.*

Figure 2: I would like to see the percentiles for each single month. This would provide also information about year-to-year variability.

*Figure 2 in the submitted version of the manuscript depicts percentile distributions based on data binned into each month over the duration measurement period. These results were interpreted to evaluate seasonal variability. While we agree with the reviewer that it would be interesting to evaluate year-to-year variability, the limited duration of the measurement period , 2.5 years, is insufficient for a reliable analysis of this nature.*

Page 6, line 189: please provide references. Possible reduced domestic emissions related to less domestic heating?

*We have added the references and clarified the text. It reads as follows:*

*"We infer that the significantly lower concentrations during the monsoon reflect the influences of synoptic easterly airflow that transports cleaner marine air mass over the region, reduced agricultural residue burning (Sarangi et al., 2014), and more efficient removal via wet deposition (Dumka et al., 2010)."*

Page 6, line 197: "Seasonal variability. . ..broad regional pattern". I do not agree. In the IGP, BC is maximized during winter months (December- January), while in Himalayas (and also at your station) the values are higher during pre-monsoon! (see also ).

*We agree with the reviewer and recognize that BC peaks during Dec-Jan in the IGP and during March- May in the Himalayan sites. We have clarified the language in section 3.1 as follows:*

*"Similar seasonal variability in BC concentration is evident across the IGP from urban to remote locations. For example, high concentrations of BC (~1.48 to 1.99 µg m-3) have been reported in near-surface air across the IGP as well as in layers of the atmosphere at ~900 m asl and ~1200 m asl during the post-monsoon over Northern India (Tripathi et al, 2005; 2007). Sreekanth et al (2007) reported BC concentrations in Vishkhapatnam, in eastern India, to be 8.01 µg m-3 in pre-monsoon and 1.67 µg m-3 during monsoon while Ramchandran et al (2007) observed BC concentrations in Ahmedabad, western India, of 0.8 µg m-3 during the monsoon in July to 5 µg m-3 during the post monsoon in January. Similar seasonable variability has also been reported in the high Himalaya. For example, the Nepal Climate Observatory-Pyramid (NCO-P) station at the 5079 m asl in the Himalaya has also shown high seasonal differences for BC (0.444 (±0.443) µ gm-3 during pre-monsoon and 0.064 (±0.101) µ gm-3 during monsoon season and ozone concentrations 61 (±9) ppbv during pre-monsoon season and 39 (±10) ppbv during monsoon (Cristofanelli et al., 2010, Marinoni et al., 2013) (Supplementary Table 2). Our results indicate that seasonable variability in BC and O3 within the KGV and presumably other deep Himalayan valleys is coupled with these larger regional-scale patterns."*

Page 6, line 209: "These differences in. . .". Not clear: what differences?

*This point is addressed in our response to the preceding comment.*

Page 6, line 213: The works by Ratikainen et al., 2014 AtmosEnv can be cited here

*We have added a citation to this study in the revised version.*

Section 3.2: this discussion is mainly qualitative. No statistical analysis have been applied and it is difficult to discern if the observed features are statistically significant. I suggest to add a line describing the mean average values with statistical confidence level (this would help in understand if the observed peak and minima are robust features). I would add to these plots the

analogous for wind direction and speed to clearly correlate wind regime with O3 and eqBc variability. In any case, the results are based just on the analysis of 3 single months of observations. A comment for taking into account the possible intra-seasonal and year-to-year variability should be added. Your measurement period is 2.5 year-long. Why you did not use all the available data?

*Figure 2 of the original version of the manuscript (Figure 3 of the revised version) depicts monthly percentile distributions for both BC and $O_3$ over the entire measurement period. As indicated in the Section 3.2 (lines 190 to 192 of the original manuscript), "representative months from each season, April 2013 (pre-monsoon), July 2015 (monsoon) and November 2014 (post-monsoon), were selected based on data availability and quality to evaluate aspects of temporal variability in more detail." Normalize diurnal variability in concentrations of O3 and BC during these months is depicted in Figure 3 of the original version (Figure 4 of the revised version). The role of wind in the transport is discussed in section 3.4. As mentioned in response to a previous comment, a 2.5-year data record is insufficient to reliable characterize inter-annual variability particularly given the data gaps in our study.*

*In addition, we have added results from non-parametric statistical analysis that shows differences between up-valley and down-valley concentration and flux. For this analysis we have used all days during the measurement period when 24 hour data for all parameters measured were available. As the data is not normally distributed a mean average value, as suggested, are not appropriate to evaluate significant differences day and night.*

Page 7, line 219: "peaked in the early afternoon". I would say "at noon"! This can be an hint for local photochemical production. . .

*The peak is too broad to assign specific time of the day. For this reason, we prefer to retain use of the term "early afternoon".*

Page 7, line 220. "Finally,. . .0 to 1 (Fig. 3)", I cannot be able to understand this sentence. . .Maybe you would suggest that diurnal variability account for the most part of the overall O3 variability? In the case, this is a point that should be better stressed. Can you quantify it?

*We have clarified this sentence to read;" In contrast, based on median values during all three periods, BC concentrations increased rapidly in the early morning, decreased during late morning, and then rose through the afternoon and early evening hours (Fig. 5)."*

Page 7, line 223: "increased rapidly following sunrise". . .because later in the manuscript, you suggested that this peak can be related to local emissions, I would change with "increased rapidly in the early morning".

*We have made the recommended change.*

Page 7, line 225-229: Again, this sentence is not clear to me. See the same comment for ozone.

*We have clarified the text to read; "Notably, BC concentrations showed lower variability relative to that of O₃, particularly during pre-monsoon periods. These skewed distributions reflect infrequent periods of relatively high BC concentrations during all three seasons."*

Page 7, line 230: you should also mention air-mass transport. At a remote site, if local emissions are really negligible (I'm not totally convinced about this for your site, see your following sentence about eqBC), I would expect that the contribution by transport is the most important one!

*We have provided further clarification, the text now reads:*

*"Several factors contributed to differences in the diurnal variability of O₃ and BC. These include diurnal variability in emissions of BC versus O₃ precursors and/or production in source regions followed by regional transport, diurnal variability in the photochemical chemical production and destruction of O₃, and contributions of O3 from non-combustion sources. O₃ is produced photochemically and is lost via deposition to surfaces and chemical reactions. In contrast, BC is a primary emission product of combustion that may originate from both local and distant sources."*

*We have also added text as further evidence that these emissions are not local. (See response to following comment)*

Page 7, line 236: the secondary peak (in the evening from 19 to 21) is visible only during the post-monsoon. Please comment. Does this peak be related with domestic emissions (e.g. domestic cooking or heating)?

*The contribution from local pollutants dominate morning peaks while the influx of pollutants after the onset of up-valley flows suggest long-range transport. The absence of secondary peak during monsoon season supports the argument. The peaks are present both during pre-monsoon and post-monsoon seasons (old figure 6). This has been clarified in the text to read;*

*"The early morning peak during all three seasons suggests probable contributions from the local combustion of biofuels for cooking and heating, which are most prevalent during early morning. The secondary peak in the afternoon and early evening occur when the local anthropogenic sources are at minimum in the KGV."*

Page 7, line 239: "Up-valley. . .Alpine pumping". As mentioned before, many works in Himalayas investigated the role of valleys as channel of anthropogenic pollution. Please consider them and comment your results as a function of these previous works.

*We have included additional references to recent papers that have investigated pollution transport in the Himalaya throughout the manuscript. Some of these studies have examined this pathway through satellite imagery and remote sensing. Others have made field-based measurements in the IGP and TP. However, as stated in the manuscript, ours is the first study to directly measure air pollution transport within a trans-Himalayan valley. We have also added a table that compares our results with results from Bonasoni (2010)*

Section 3.3: the expected outcome from this Section is not clear. Why did you show just 6 days of data at JSM_2, when more than two years of meteorological data are available at the "core" site where also O3 and eqBC data were available? You must show these data! Moreover, if I'm not wrong, JSM_2 is located 1000 m above the "core" site. Thus, which is the goal of showing these data?

*In the original manuscript, the station names at Jomsom were in misplaced. These errors have been corrected and we apologize for any related the confusion.*

*We have updated the old wind rose figure with wind roses for JSM_2 for all seasons (new figure 6a) binned every three hours for the duration of the measurement period. The wind roses show the seasonality in wind direction and magnitude of wind speed at Jomsom.*

Section 3.4: Legend is missed in Figure 6. I suspect that blue dots represent O3 but you have to add a legend! Basically, this section repeat the same concept about diurnal variability already reported by Section 3.2. . .

*The legend has been added. Section 3.2 describes the diurnal variability in the BC and O3 concentrations. Section 3.3 describes the wind pattern in the valley. Building on these lines of evidence, section 3.4 describes how BC and O3 concentration variability correlates with wind variability. It is important to state this link explicitly before we describe the anomalies we observe in diurnal concentration (section 3.5).*

Page 8, line 271: "This peak occurred about an (one) hour later during the post-monsoon period". Looking at Figure 3, this seems not true! eqBc peak at 8:00 AM during all the seasons. However, it is important to evaluate the robustness and origin of this peak. Looking at the eqBc time series reported in Figure 6, it looks that the early morning peak is related to "spiky" observations, very likely related to local emissions. This is particularly evident during post-monsoon, when these "spikes" where observed during the diurnal minima of eqBC. This feature can be of a certain interest to evaluate the local emissions to the "pristine" Himalayan environment, but I would neglect it for the analysis of transport processes affecting O3 and eqBC variability.

*Please refer to the response earlier to the comment about secondary peaks.*

Page 8, line 275: ". . ..decreasing concentration with increasing wind speeds are consistent with expectation based on dilution". I think that the decrease on eqBC observed during midday can be associated not only to dilution in a more developed PBL of local emissions, but also to the fact that air-masses reaching the site from the lower valley are still not enriched in pollutant. . .Indeed, you observed eqBC (and O3) increase in the afternoon/evening (when both PBL height decrease and air-masses richer in pollution could reach the site from longer distance). You should roughly evaluate the distance of eqBC emissions by analysing wind speed at the measurement sites. . .

> *We have roughly calculated the maximum distance from which the secondary peak emissions could have originated during May 2015. We estimate that emissions could have traveled from as far as 70km away from Lete. This estimate was done by calculating the time it would take for the secondary peak occurring at Jomsom (1900 LST) to travel from Lete (1810 LST) to Marpha (1850 LST) to Jomsom. We then determined the difference in time (8 hours) between when emissions started to increase after the morning peak at Jomsom and the occurrence of the secondary peak at Jomsom. Finally, we used this 8 hour estimate combined with wind speed measurements at Lete to calculate the maximum distance from Lete from which secondary peak emissions could have originated. This poorly constrained estimate does not account for other factors that influence pollutant transport and its reliability cannot be evaluated based on objective criteria. Consequently, we choose not to include it in our paper.*

Page 8, line 285. This detailed description of local wind regime needs a more detailed map on Figure 1!

> *The revised manuscript includes a more detailed map (new figure 2).*

Page 9, line 294: "Nocturnal decupling of the boundary layer preserves the concentration of. . .". Not clear: what do you mean with "decoupling PBL"? Decoupling from what?

> **Author:** *We have removed the sentence.*

Section 3.4: I assume that this Section should be "Evidence of LONG-RANGE transport episodes. . .". In this section you discuss three typical regimes of O3 and eqBC variability. However, I would like to see a more detailed description of the main features of each single regime (basically how are you able to distinguish among them?) and a systematic assessment of their occurrence and impact on eqBC and O3 variability E.g. which is the frequency of occurrences of these regimes on a seasonal basis? Are you able to objectively identify (by some

selection criteria) the occurrence of that regimes? Can you able to compare your results with previous studies? (e.g. Putero et al., Environ. Poll. 2013)

*We have changed the section title to "Evidence of regional transport episodes". We also updated the figure with wind data for the episodes shown as examples (new figure 8); have quantified the frequency of each transport pattern and included all the transport pattern observed in supplementary table ..*

Page 9, line 304: actually you did not show any evidence of transport to TP! I think this is just a (reasonable) speculation. Maybe you can discuss this possibility in the conclusions Section.

*We have added transport to TP in the conclusion as a potential transport to the TP.*

Page 9, line 310: I think that the relationship between eqBc , O3 and fire emissions in the IGB is only qualitative and deserve more analyses. At least a correlation between the temporal variation of the fire number in the IGP and the eqBc and O3 at the measurement site must be showed. Moreover, air-mass transport analysis (i.e. backtrajectories or dispersion plume) corroborating the transport towards the measurement site region should be showed (the same is valid for case B). For case B and C, eqBc is maximized during night-time, when down-valley winds are expected in a mountain valley and cleaner air-masses from upper layers should be present (add the behaviors of wd and ws!). Please, comment. Moreover, the different co-variability (correlation) between O3 and eqBC should be better investigated and commented. During case B, the diurnal variability of O3 and eqBC appeared to be minimized. Can this indicate the role of far (fires?) emissions instead of regional/local ones? The increase of daytime minima value of eqBc and O3, would indicate a build-up of pollution. Is this build-up only limited to the valley or did it extend to the foothills? Maybe time series of satellite MODIS data can help. . .

*The authors feel that back trajectories trying to relate surface sources to valley transport are unreliable in complex terrain so we used HYSPLIT back trajectories from the mouth of the valley in conjunction with the MODIS imagery. The back trajectories show transport of air mass from are with high biomass/ forest fires activity and/or haze over the IGP during the regional transport period examples. It supports the hypothesized source for the high concentration during regional transport episode. Dispersion models are beyond the scope of this paper.*

Section 3.4.2 is confused and not provides important information: thus, it can be skipped. STE discussion at this point is a little bit out of the scope of the paper. A rigorous assessment of STE contribution to eqBC and O3 variability would deserve a specific investigation. Moreover, I would note that the focus on the monsoon season is of limited interest, since it is well assessed

that summer monsoon is the season during which transport of stratospheric air-masses to the lower troposphere is minimized over the Himalayan region (see e.g. Putero et al., 2016, Ohja et al., , 2016).

> *We have removed discussion of STE from the paper.*

The conclusions are not robust. Actually you did not demonstrate the transport of pollutant from IGP to TP but just the transport of pollution up to the valley. Your main results (seasonal and diurnal O3 and eqBC variability, influence of open fires) should be better commented in the framework of the most recent studies on the topic.

> *In response to the reviewer's comment regarding transport to the TP, we have revised the text to indicate that that our observational evidence suggests (but does not conclusively demonstrate) such a connection.*

**Referee #2**

The manuscript reports data that is collected recently over a high altitude site and does not bring in any new insights or results. As the authors write, many results from the same group have been published recently.

> *Numerous studies have looked at the possibility of the transport of pollutants from the IGP to the Himalayan foothills (Pant et al., 2006; Dumka et al., 2008; Komppula et al., 2009; Hyvärinen et al., 2009, 2010; Ram et al., 2010; Brun et al., 2011; Gautam et al., 2011; Srivastava et al., 2012b).  In addition, the obstruction of flow caused by the high Himalaya intensifies the effect of pollution over the IGP that are visible in satellite imagery especially during pre-monsoon seasons (Singh et al., 2004; Sarkar et al., 2006; Bollasina et al., 2008; Gautam et al., 2009; Dey and Di Girolamo, 2010). Other studies show that pollutants, with similar regional sources, have the potential to reach not only the foothills of the Himalaya but also higher elevations (Decesari et al., 2010; Marinoni et al., 2010; Sellegri et al., 2010; Gobbi et al., 2010). Though there is evidence of existence of similar source pollutants, both regional and local, in the foothills and the higher altitude sites (Raatikainen et al., 2016, Raatikainen et al., 2014, Srivastava et al., 2012,), observational evidence of mechanisms and pathways facilitating such transport is lacking. Our data  fills this  gap by characterizing  the role of the wind system within a deep Himalayan valley in transporting pollutants from the IGP to the high mountains.*

> *These data have not been previously published.*

What is the relevance of connecting ozone and BC is not clear. Many aspects are very loosely dealt with and mentioned in passing. References are missing.

*The rationale for investigating BC and ozone is explained in detail in the introduction. Briefly, both species are short-lived climate forcers (SLCFs) that originate from combustions sources over the IGP. We measured these species in conjunction with corresponding meteorological conditions and satellite imagery to understand the importance of deep valleys in their transport to the high Himalaya. Impacts on ice-albedo and temperature contributes to warming, which accelerated melting of glaciers with potential negative consequences for almost a billion people in the surrounding watersheds. Mitigation of SLCF emissions has the potential to slow the rate of future climate change. As indicated in our responses to comments by Reviewer 1, we have added references relevant to the analysis.*

Lines 285 - how this phenomenon can occur? What about lifetime of BC aerosols?

*As indicated above in our response to a similar comment by Reviewer 1, we roughly estimated the transport time required for an air mass from Lete to Jomsom (approximately 50 minutes) and the distance of the source from Lete (approximately 70km). Given the lifetime of BC and in the absence of substantial wet or dry deposition, emissions from IGP could reach Lete and up the KGV under suitable synoptic and local weather.*

Conclusion - with the limited scope of the study the conclusions are far fetched.

*While the reviewer was not specific regarding the nature of his/her concern, we have exercised greater caution in the statements regarding transport of pollutions from the IGP to the TP and revised the conclusion section. It now reads,*

*"This study provides in-situ observational evidence of the role of a major Himalayan valley as important pathway for transporting air pollutants from the IGP to the higher Himalaya. We found that:*

*•        Concentrations of BC and O3 in the KGV exhibited systematic diurnal and seasonal variability. The diurnal pattern of BC concentrations during the pre- and post-monsoon seasons were modulated by the pulsed nature of up-valley and down-valley flows. Seasonally, pre-monsoon BC concentrations of BC were higher than in post-monsoon season.*

*•        The morning and afternoon peaks in the post monsoon season was more pronounced than those of pre-monsoon season likely due to the relatively lower wind speeds during post-monsoon.*

- *When compared to a high elevation site, NCO-P CNR in the Himalaya, JSM_STA consistently showed higher BC concentrations for all seasons whereas the corresponding O3 concentrations were higher at NCO-P CNR.*

- *Significant positive up-valley fluxes of BC were measured during all seasons.*

- *During episodes of regional pollution over the IGP, relatively higher concentrations of BC and O3 were also measured in the KGV.*

*Further studies are needed to understand the vertical and horizontal distribution of particulate matter and ozone in the Himalayan region, and their impact on the radiative budget, the ASM and climate. Investigations using sondes, LiDar and air-borne measurements could help characterize the stratification of the vertical air masses."*

[revised manuscript text omitted]
., 2012). Other studies show that pollutants have the potential to reach not only the foothills of the Himalaya but also higher elevations (Bonasoni et al., 2010; Decesari et al., 2010; Marinoni et al., 2010). In addition, studies have shown the obstruction of flow caused by the high Himalaya which intensifies the effect of pollution

150 over the IGP that are visible in satellite imagery especially during pre-monsoon seasons (Singh et al., 2004; Dey and Di Girolamo, 2010; Gautam et al., 2011). Though there is evidence of existence of similar source pollutants, both regional and local, in the foothills and the higher altitude sites (Raatikainen et al., 2017; Raatikainen et al., 2014; Srivastava et al., 2012,), observational evidence of mechanisms and pathways facilitating such transport via Himalayan valleys is lacking. Our data fills

[revised manuscript text omitted]

**3.3 Evolution of local wind system in the KGV**

Measurements from JSM_2 show the diurnal evolution of wind in each season. All data collected were used to analyze the diurnal pattern of wind at Jomsom (Fig. 6). The wind roses illustrate the temporal evolution of up- and down-valley flows at JSM_2 for each season (Fig. 6). At JSM_2, up-valley flows are southwesterly and dominant during daytime with peak velocities above 15 m s$^{-1}$ between about 0900 LT to 1800 LT. Wind velocities decreased substantially after 1800 LT, with variable wind direction until midnight then northeasterly winds are common during pre- and post-monsoon seasons (Fig. 6a and 6c)). The wind pattern during monsoon was strongly influenced by the monsoon anticyclone, this observation is in agreement with wind direction from other Himalayan valleys (Bonasoni et al., 2010; Ueno et al., 2008) (Fig. 6b) Although wind velocities at JSM_2 varied somewhat over the year, non-monsoon months exhibited similar diurnal patterns that evolved seasonally as a function of sunrise and sunset (Fig. 6) unlike in the monsoon season. As discussed below, this alternating pattern in wind direction from strong daytime flows to weak nighttime flows during dry months results in a net transport of pollutants up the valley.

Theodolite observations at different locations along the KGV in 1998 show minor shifts - less than 45° - in wind direction and less than 2 m s$^{-1}$ in wind speed in the lower 1000 m above the surface during daytime (Egger et.al 2000). Based on the average daytime wind speed, the valley can be partitioned into three regions: the entrance, core, and exit (average wind speeds range from 5 to 10 m s$^{-1}$, 8 to 18 m s$^{-1}$, and less than 5 m s$^{-1}$ respectively) (Egger et al 2000). The strongest winds within the core region are most prevalent in the lower 1000 to 1500 m of the boundary layer within the valley (Egger et al, 2000; Zängl et al., 2000, Egger et al., 2002).  Our measurements at the four AWS stations on the valley floor illustrate the evolution of surface wind velocities along the length of the KGV (Supplementary Figure 1). In addition, comparison between measurements at JSM_1 between 1st and 14th May 2015 and the longer-term record JSM_2 for same period (Fig. 6 and Supplementary Figure 1) provide information regarding vertical variability. Velocities at the higher elevation site of JSM_2 were about 5 m s$^{-1}$ greater than those near the valley floor during the day and 3 m s$^{-1}$ stronger at nighttime but with similar diurnal cycles. While there is a relatively stronger northeasterly wind at JSM_2 from 0300 to 0900 LST in comparison to JSM_1. Airflow along the valley is driven by gradients in temperature and pressure between the entrance and the exit regions of the valley (Egger, 2000). Wind speed along the valley floor peaked within the core of the valley at MPH, JSM_1 and JSM_2 and were lower in the entrance (LET) and exit (EKL) regions (Supplementary Figure 1). The duration of strong wind speeds within the valley

[revised manuscript text omitted]

We identified three common patterns in the BC profile at Jomsom during the transport episodes. For most of the transport episodes were associated with observational evidence from satellite imagery of emission sources within the region including haze over IGP, agricultural and biomass burning in Punjab regions of India and Pakistan, and forest fires in the foothills of the Himalaya in northern India or

southern Nepal (Supplement Table 3). The 90th percentile for each year 2013, 2014 and 2015 were 1.53 µg m$^{-3}$, 1.60 µg m$^{-3}$ and 1.47 µg m$^{-3}$ respectively. The year 2015 only included January through July data. We partition these episodes into three characteristic patterns based on relative variability. Pattern A was characterized as a fluctuating daily maxima in BC with peaks that repeatedly exceeded the 90th percentile but with daily relatively lower minima (Fig. 8[Ia]). Pattern B was characterized as regional transport periods when a buildup of BC concentration was seen over the period without a relatively low daily minima but peak concentrations over the 90th percentile (Figure 8b). While Pattern C corresponded to the periods when the BC concentration exhibited both pattern A and B during a single regional episode (Figure 8c). A total of 31 regional episodes were identified from January 2013 till July 2015, 50% of which were pattern A, 30% Pattern B and 20% pattern C. The wind speeds at Jomsom during these transport episodes exhibited diurnal variability similar to those during other periods (Fig. 8 [II]). We initiated the Hybrid Single-Particle Lagrangian Integrated Trajectory model (HYSPLIT), developed by NOAA's Air Resources Laboratory, from 300 m, 500 m and 1000 m for 72 hours runtime for each of the example episodes (Stein et al., 2015). HYSPLIT trajectories show that the transport of air mass from the above source regions during that period (Fig. 9).

During the regional transport period in November 2014 (Pattern A), average daily BC concentration was 1.29 µg m$^{-3}$ which is over the 75th percentile (0.88 µg m$^{-3}$) of the BC concentration for the measurement duration. The maximum daily concentration during the period was 3.04 µg m$^{-3}$. However, the corresponding average O$_3$ was only 28.07 ppbv, slightly below the average (29.48 ppbv) for the entire data set (Figure 8 [Ia]). The diurnal wind pattern in the KGV was conserved during pattern A (Figure 8 [IIa]). The MODIS fire data and HYSPLIT back trajectories showed extensive haze and fire events during this period of regional transport (Figure 9). We infer that transport of BC emitted from agricultural burning and wildfires during this period contributed to the high concentration measured with the KGV.

The mean BC concentration during pattern B, one example of which occurred in May 2014, was 1.77 µg m$^{-3}$ (Figure 8[Ib]). It was above the 90th percentile (1.49 µg m$^{-3}$) for entire measurement period while O$_3$ concentrations were at 49.71 ppbv, slightly below the 90th percentile (52.9 ppbv). The wind pattern in the KGV exhibited diurnal flow patterns but with longer period of up-valley flows compared to pattern A (Fig. 8 [IIb]). During this period MODIS imagery and HYSPLIT back trajectories revealed widespread burning in the Himalayan foothills and the Punjab region of India (Figure 9).

One of the Pattern C- type of transport episodes was identified in May 2013, when the average concentration was well above the 90th percentile for both BC (2.09 µg m$^{-3}$) and O$_3$ (57.49 ppbv) (Fig. 8 [Ic]). Diurnal wind patterns were similar to those of pattern B with extended duration of up-valley flows (Fig. 8 [IIc]). MODIS and HYSPLIT back trajectories revealed extensive agricultural burning in the Punjab region of India and the southern plains of Nepal during this period (Figure 9).

[revised manuscript text omitted]

- When compared to a high elevation site, NCO-P CNR in the Himalaya, JSM_STA consistently showed higher BC concentrations for all seasons whereas the corresponding $O_3$ concentrations were higher at NCO-P CNR.

- Significant positive up-valley fluxes of BC were measured during all seasons.

- During episodes of regional pollution over the IGP, relatively higher concentrations of BC and $O_3$ were also measured in the KGV.

- Further studies are needed to understand the vertical and horizontal distribution of particulate matter and ozone in the Himalayan region, and their impact on the radiative budget, the ASM and climate. Investigations using sondes, LiDar and air-borne measurements could help characterize the stratification of the vertical air masses.

- .

[revised manuscript text omitted]

Hegde P., Pant, P., Naja, M., Dumka, U. C., and Sagar, R.: South Asian dust episode in June 2006: Aerosol observations in the central Himalayas, *Geophys. Res. Lett.*, 34, L23802, doi:10.1029/2007GL030692, 2007.

705   Henne, S., Furger, M., Nyeki, S., Steinbacher, M., Neininger, B., de Wekker, S.F.J, Dommen, J., Spichtinger, N., A. Stohl, A. and Prevót A. S. H.: Quantification of topographic venting of

boundary layer air to the free troposphere, *Atmos. Chem. Phys.*, 4, 497–509, 2004.

Hindman, E. E. and Upadhyay, B. P.: Air pollution transport in the Himalayas of Nepal and Tibet during the 1995–1996 dry season, *Atmos. Environ.*, 36, 727–739, 2002.

[revised manuscript text omitted]

MOVED FROM THE MANUSCRIPTSupplementary Figure 1 (a) Wind rose for data from May 8 to 14, 2015 binned into 3-hour increments depiciting diurnal evolution in wind speed and direction at JSM_1. (b) Corresponding diurnal variability in wind speed based on average values over the same period at LET, MPH, JSM_1 and EKL.

NEW Supplementary Table 2 Mean, median and percentage distribution for BC concentration and flux for up-valley and down-valley flows. Net daily flux for each season and its percentage distribution.

| Season | Variable | Northeasterly (down-valley) | | | | Southwesterly (up-valley) | | | | Difference | | | |
|---|---|---|---|---|---|---|---|---|---|---|---|---|---|
| | | Mean | Median | 25th Percentile | 75th percentile | Mean | Median | 25th Percentile | 75th percentile | Mean | Median | 25th percentile | 75th percentile |
| Pre-monsoon | BC concentration | 1.060 | 0.793 | 0.464 | 1.420 | 0.980 | 0.713 | 0.383 | 1.307 | 0.080 | 0.080 | 0.081 | 0.113 |
| | BC flux | 14.088 | 9.134 | 4.573 | 18.160 | 38.030 | 26.177 | 12.756 | 52.812 | -23.942 | -17.043 | -8.183 | -34.652 |
| | Net flux* | 0.750 | 0.510 | 0.260 | 1.143 | | | | | | | | |
| Monsoon | BC concentration | 0.375 | 0.280 | 0.150 | 0.430 | 0.179 | 0.127 | 0.070 | 0.227 | 0.196 | 0.153 | 0.080 | 0.203 |
| | BC flux | 3.980 | 2.636 | 1.182 | 4.555 | 5.984 | 4.063 | 2.240 | 7.592 | -2.004 | -1.427 | -1.058 | -3.037 |
| | Net flux* | 0.220 | 0.147 | 0.108 | 0.271 | | | | | | | | |
| Post-monsoon | BC concentration | 0.800 | 0.653 | 0.327 | 1.237 | 0.730 | 0.600 | 0.070 | 1.110 | 0.070 | 0.053 | 0.257 | 0.127 |
| | BC flux | 10.436 | 6.700 | 3.184 | 13.874 | 27.020 | 19.530 | 2.240 | 39.300 | -16.584 | -12.830 | 0.944 | -25.426 |
| | Net flux* | 0.372 | 0.275 | 0.980 | 0.604 | | | | | | | | |

NEW Supplementary Table 3. List of enhanced BC episodes observed at JSM_STA and the concurring regional sources from MODIS (* Data is from January-July)

| MONTH | EPISODE LENGTH | EPISODE TYPE | SOURCE LOCATION |
|---|---|---|---|
| **2013 (90th Percentile = 1.53)** | | | |
| MONTH | EPISODE LENGTH | EPISODE TYPE | SOURCE LOCATION |
| Jan | 6th-15th | C | Haze |
| | 28th-31st | C | fire in the Punjab region of Pakistan |
| Feb | Jan (contd.)- Feb 2nd | C | Haze |
| Mar | 1st-3rd | A | Haze along with fires west of Nepal in northern India |
| | 6th-9th | A | fire in the Punjab region of Pakistan |
| | 11th-13th | B | fire in the Punjab region of Pakistan |
| | 18th-27th | B | fire in the Punjab region of Pakistan and scattered fire in Northern India, west of Nepal |
| Apr | 6th-10th | B | fire in the Punjab region of Pakistan and in central and western Nepal |
| | 12th-14th | B | fire in the Punjab region of Pakistan and fire in central Nepal |
| | 27th-30th | A | fire in the Punjab region of Pakistan and in Northern India-west of Nepal; Haze |
| May | 3rd-11th | B | fire in the punjab region of India (over 100 events) and in Pakistan |
| Oct | 28th-30th | A | fire in the punjab region of India and Pakistan and Haze |
| Nov | 1st-5th | A | fire in the punjab region of India and Haze |
| | 23rd-30th | A | Haze |
| Dec | 17th-24th | C | Haze |
| **2014 (90th Percentile=1.60)** | | | |
| MONTH | EPISODE LENGTH | EPISODE TYPE | SOURCE LOCATION |
| Jan | 13th-15th | A | Cloud cover |
| Feb | Jan31st-Feb 1st | A | high cloud cover |
| Mar | 14th-16th | A | fire in Punjab region of Pakistan |
| Apr | 5th-8th | B | fire in central and western Nepal and Haze |
| | 10th-30th | C | fire events in Punjab, India and Pakistan |
| May | Apr 10th-May 1st | C | fire in western Nepal, central India and Punjab, Pakistan |
| | 8th-13th | B | fire in western Nepal and over 150 fire events in the Punjab region of India and Pakistan |
| | 17th-23rd | C | fire in India-west of Nepal- and over 100 fire events in the Punjab region of India and Pakistan |
| Jun | 5th-8th | B | fire in India-west of Nepal (Uttarakhand-India) |
| | 10th-17th | B | fire in the foothills of the Himalaya (western Nepal and northern India-west of Nepal |
| Nov | 15th-30th | A | Haze and fire events in India |
| Dec | Nov 15th-Dec 8th | A | Haze and a few fire events in Punjab, Pakistan |
| **2015 (90th Percentile=1.47)*** | | | |
| MONTH | EPISODE LENGTH | EPISODE TYPE | SOURCE LOCATION |
| Jan | 15th-17th | A | Cloudy/indeterminant |
| | 21st-25th | A | Cloudy/indeterminant |
| Feb | 21st-26th | A | Cloudy/indeterminant |
| May | 6th-9th | A | Extensive butning (over 200 fire events) in Punjab region of Pakistan and India |
| | 18th-26th | A | fire events in the western region of Nepal and Northen India- southwest of Nepal |
| | 29th-31st | B | Fire in northern India-west of Nepal (Uttarakhand) and western Nepal |
| Jun | 6th-12th | B | Fire in northern India-west of Nepal (Uttarakhand) and western Nepal |

---

## Author Response (AR2)

Dear Editor,

Thank you to you and the reviewers for your valuable comments regarding our study. Our responses below and the improvements that we have made to the manuscript have now thoroughly addressed the remaining concerns of the reviewers. Specifically we have: 1) expanded the methods section to detail our treatment of the data and our calculations of BC mass transport, 2) removed more speculative aspects of the study (including the examination of back trajectories), 3) included greater description of the limitations of our data and the conclusions that we draw from it, and 4) ensured that the grammar and structure are of suitable quality for ACP. In all, these changes have helped to greatly improved our manuscript which we feel is now suitable for publication in ACP.

Thank you for your continued consideration,

Shradda Dhungel*

*shradda@virginia.edu

Reviewer #1

This is my second review of this paper. In respect to the first submission, the authors added some new elaborations/analyses and accepted some of my previous indications. The paper is improved from the original submission (as an instance the authors improved the motivation of the study and the manuscript organization) but more work is still necessary before it can be considered suitable for publications by ACP. Indeed, the paper still suffers of major deficiencies (both technical and scientific). It still appears somewhat confused and still difficult to follow. Besides the presentation of O3 and BC measurements at a new Himalayan location, no new information in respect to previous similar works have been provided (the main outcome of this paper is that the valley represent a "channel" for pollution to be transported at high altitudes). However, as I reported in my previous review, in the Himalaya region a need for measurements and scientific evidences still is needed, and the observations from KGV can provide useful information for (i) confirm the role of Himalaya valley as "preferred" way for the anthropogenic pollution to be transported up to high Himalaya, (ii) support the hypothesis that pollution from IGP and Himalaya foothills can be transported across mountain ridge to Tibetan Plateau or vented to the free troposphere, (iii) assess the contribution of open fires (or other emissions) to the occurrence of the ABC in South Asia/Himalayas region.

We thank the reviewer for his/her continued feedback on our manuscript. The main objective of this study is to present and describe new observational evidence of pollution transport from the Indo-Gangetic Plan through Himalayan mountain valleys to the higher Himalaya and Tibetan Plateau. We have clarified this main objective in the text and also clearly stated that our assessment of BC flux is preliminary and that future studies should seek to quantify flux and pollutant sources with greater depth and rigor. We have also removed any assessment of pollutant source regions as performed with back trajectories. We have also substantially reorganized the manuscript (particularly regarding the methods and results/discussion

sections). Below can be found our responses to specific comments made by reviewer #1, all of which have helped to greatly improve our manuscript.

From a technical point of view, the paper need a strong language revisions and the quality of some figures must be improved.
We have made a thorough, global review of the language for grammar and terminology and leave this technical aspect at the discretion of the editor. While we could not be certain to what specifically the reviewer was referring regarding figure quality, we have made several improvements to improve clarity and readability (see Figures 2 and 3).

For these reasons, I cannot recommend publication but reconsideration after major revisions.

MAJOR POINTS
In respect to my previous review you mentioned that "winter" was not considered in your paper because is "not relevant". Please, better argument this point: January and February are not winter months?
In the methods section, we define each season that we consider based on wet or dry season. The seasons are thus defined a pre-monsoon, monsoon and post-monsoon season. Winter months fall within the post-monsoon season.

Line 273: as already reported in my previous review, this is not true. The seasonal cycles ARE DIFFERENT between IGP and Himalaya foothills and Himalayas. Your BC peak is in pre-monsoon. While over the plains or foothills the peak is during winter (see also Putero et al., ACP, 2015 same special issue). The seasonal cycles are different not similar.
We have changed the text to more accurately reflect the differences in timing. The added text now reads "The post-monsoon timing of peak BC concentrations observed in previous studies performed in the IGP and Himalayan foothills (see e.g., Tripathi et al, 2005; 2007; Ramchandran et al., 2007; Putero et al, 2015) differs from our observations in the KGV, where we see heightened BC during the pre-monsoon season (Fig. 5). These findings are generally in agreement with other high altitude observations. For example, the Nepal Climate Observatory-Pyramid (NCO-P) station at the 5079 m asl in the Himalaya has also shown high seasonal differences for BC and $O_3$ between pre-monsoon (0.444 (±0.443) μg BC m$^{-3}$; 61 (±9) ppbv $O_3$) and monsoon (0.064 (±0.101) μg m$^{-3}$; 39 (±10) ppbv $O_3$) (Cristofanelli et al., 2010, Marinoni et al., 2013) (Table 2). Our results therefore indicate a lagged peak in BC and $O_3$ within the KGV and presumably other deep Himalayan valleys, as compared to sites within the IGP.".

Line 347-366: this discussion is out of the scope of the paper. Please remove.
We have removed this text.

Line 430: to provide a more robust assessment about the degree of mixing within the valley you should calculate the gradient of potential temperature between the two AWS...
We have created a plot of potential temperature to further support our argument. See Supplementary figure 2.

"Because….estimate". On which basis can you affirm that BC can be vented above 800 m agl.

Provide references.

We have now provided photographic evidence that the higher elevation JSM_2 was within the polluted layer (Supplementary Figure 1).

Section 3.4: you should better discuss the different relationship between BC and O3 for the different class of events, not only limiting to describe the observed behavior but also speculating about the reasons leading to the different co-variabilities. As an instance, why during the Pattern A no O3 enhancement is observed? On the other side for Pattern B the O3 is highest when BC is low. A mixed situations occur for pattern C. Please comment and provide explanations. Figure 8: please in the caption provide explanation for the dotted lines and blue/orange dots.

We prefer not to speculate about the reasons leading to different patterns as this would require a more robust analysis than is pursued in the present study. We have added a brief statement to indicate some of the factors that might contribute to these differences: "Different atmospheric lifetimes, chemical reactivity, source location, and sinks of BC and $O_3$ all may contribute to these differences in BC and $O_3$ concentrations detected in the valley." We have also improved the figure caption to indicate what the dashed lines and blue and orange dots represent.

The Conclusion Section looks like a simple summary/highlights of the evidences reported long the paper.

Once again: you should discuss the fact that only 2.5 years of observations are available. Thus you are not able to discuss and assess interannual variability of frequency and magnitude of observed pollution transport.

A conclusion section should indeed serve as a summary of the paper. We have modified the conclusion section and included mention of the limitations regarding interannual variability as highlighted by the reviewer.

TECHNICAL/SPECIFIC/LANGUAGE POINTS

Abstract-line 28: please revise this sentence ("particulary during preceeding the monsoon" replace with "pre-monsoon")

We have made the recommended change.

Line 182, please replace "climatologically important pollutants" with "essential climate variables (ECVs)"

We rewrote this phrase to read "two important SLCPs".

Line 192: I'm sorry but the valley orientation description is still obscure to me. What do you mean with "The valley is oriented southeasterly to northwesterly"? Maybe "The valley is oriented from SE to NW"? The same for the following sentences…

We clarified the text to read: "The general orientation of the valley is from SW (the mouth of the valley) to NE (the head of the valley) (Fig. 1)."

Figure 2: please for each site provide the start-end date of the available observations.

We have added the start and end dates of data availability.

Move Table 1 Supplementary to the main manuscript (section 2.1).
We have now moved Supplementary Table 1 to the main text.

Line 208: Again, more details about QA/QC should be reported. How the O3 sampling is executed? Do you use anti-particulate filters at the inlet lines (which type of filters?). By which frequency the instrument is calibrated? Against which standard? How the inlet line is made? Did you calculate the measurement uncertainty?
We have included greater detail about our pollutant sampling. The added text now reads: "The atmospheric observatory at Jomsom (JSM_STA) is equipped with instruments to measure BC, $O_3$, and meteorology (Figure 2). The observatory is located on the southeast corner of a plateau jutting out from an east-facing slope about 100 m above the valley floor and with no major obstructions either up or down the valley. Equivalent black carbon (hereafter referred to as BC) was measured with a Thermo Multiangle Absorption photometer (MAAP), model 5012 that uses a multi-angle photometer to analyze the modification of radiation fields – as caused by deposited particles that entered through a straight, vertical inlet line – in the forward and back hemisphere of a glass-fiber filter (GF-10). MAAP was operated at a flow rate of 20 L min$^{-1}$, measuring BC at 1-minute frequency. We note that Hyvärinen (2013) illustrates the artifact in MAAP measurements in environments with high aerosol loading with an underestimation of concentration above 9 µg m$^{-3}$. Since the median monthly concentrations for the duration of the measurement were less than 1µg m$^{-3}$ and 90$^{th}$ percentile below 2 µg m$^{-3}$ (and therefore below this threshold), MAAP corrections were not applied. $O_3$ was measured with a 2B Tech model 205 via the attenuation of ultraviolet light at 254 nm passing through a 15 cm long absorption cell fitted with quartz windows. The instrument was operated at a flow rate of 1.8 L min$^{-1}$. For instrument calibration, the BC instrument performed an automatic span and zero checks every 24 hours while zero checks on the $O_3$ instrument were performed every 7 days. Wind speed and direction were measured by an automated weather station installed on a ridge 900 m (JSM_2) above the sampling site for BC and $O_3$.".

Line 259: is not suppressed but reduced
We have made the recommended change.

Line 283: replace "seasonable" with "seasonal"
We have made the recommended change.

Line 296: please add formula to better explain you calculations. I'm still convinced that showing concentration (for BC) and mixing ratio (for O3) will provide more useful info for the audience.
Done, we added the formula for normalization in section 2.2 Data summary.

Line 319: use subscript for "3"
We have made the recommended change.

Line 319. "non-combustion sources": please explain more explicitly
It now reads, "or its precursors from non-combustion sources like stratospheric ozone and biogenic hydrocarbons from vegetation."

Line 329: something is missing in this sentence. "… are at minimum in the KGC, thus supporting the contribution of regional/long-range transport".
We have corrected this sentence.

Line 334: "in each seasons". Do you mean the single months showed by Figure 5? In the case, please, replace "pre-monsoon" with "Apr. 2013" and so on…The quality of Figure 2 is poor. Especially for post-monsoon is really difficult to read the wind roses. Please, zoom by decreasing the upper range of radius…
We have added the months that correspond with each season. However, it was unclear to us what specifically the reviewer meant by "poor quality" of Figure 2. We have modified the Figure 2 to ease its readability and believe that the figure adequately conveys its intended information. Small text at the bottom of the image is simply the result of taking image from Google Earth. As for the wind roses in Figure 5 now Figure 3, we wish to maintain consistent axes for all wind roses in order to allow comparison across time steps. We leave it to the editor to decide.

Line 392: Figure 4 appears after Figure 5 and 6 (section 3.2, 3.3). Please check and re-number figures where appropriate. Once again. Figure 4 is not useful for the paper. Each feature that you describe here can be (more easily) find looking at Figure 5 and 6. I suggest to move Figure 4 in the supplementary materials just to provide an overview of data.
We have ensured that figure numbers are consistent and ordered as they occur within the text.

Line 366: replace "one hour" with "1 hour"
We have made the recommended change.

Line 403: this is not true looking at Figure 5 where only 1 peak can be seen in late afternoon.
We have now clarified that this statement was in reference to BC concentration. The revised text now reads: "Distinct morning and afternoon peaks in BC concentration are seen in the post-monsoon season when the up-valley wind speeds are relatively weaker than in pre-monsoon season (Fig. 4 and 6)."

Line 406: replace "relatively calmer" with "weaker"
We have made the recommended change.

Line 407: BC diurnal behavior at NCOP(reported by Marinoni et al., ACP, 2010 and not by Bonasoni et al., 2010) is not bimodal but characterized by a single peak in late afternoon/evening.
We thank the reviewer for this point and have corrected the text to reflect this different behavior. The corrected text now reads: "The bimodal diurnal distribution of BC concentration in Jomsom is similar to that observed in Kathmandu (Putero et al. 2015) and unlike a singular late afternoon/evening peak seen at high elevation sites (Bonasoni et al., 2010) during non-monsoonal seasons. This illustrates that deep Himalayan valleys are susceptible to diurnal pollution similar to that of urban areas like Kathmandu. The morning peak is most likely due to local pollutants (from household and morning aircraft traffic) in Jomsom and settlements downwind of Jomsom, while the afternoon peak is associated with

long range transport. At the same time, $O_3$ exhibits a distinct minimum in the early morning with concentrations increasing towards an early afternoon peak – occurring well before BC's afternoon peak. The $O_3$ minimum in the morning further supports that the morning BC peak originates from local sources, as $O_3$ is only formed downwind of pollution sources. Further, the Jomsom station measuring BC and $O_3$ (JSM_STA) is located more than 100 m above the valley floor where the village of Jomsom sits. As such, we do not expect that local evening emissions would reach the stations at the cessation of up-valley flows and that evening drainage flows, following the valley floor would remove these local evening emissions down-valley.".

Line 418: "the distribution of BC concentrations…"..maybe " the differences of BC concentrations…"?
The revised sentence now reads: "The differences in BC concentration between up-valley and down-valley flows are statistically significant for all seasons (Figure 7a)."

Figure 7: I would say that DV fluxes must be negative…

Line 437: from Supplementary figure 1, wind speed at JSM_2 is more than 5 m/s higher than at JSM_1 (which is about 30%). Hardly you can define them "similar"… Please uniform station codes with Figure 2.
The reviewer makes a good point which we did not explain adequately in our previous version. We have acknowledged this difference and provided further justification for our approach. The text now reads: "Egger et al. (2000) used theodolite measurements to demonstrate uniform wind speeds within the bottom 1000m above the KGV floor at Jomsom and other locations. While we observed differences in wind speed of ~5 m s$^{-1}$ between the two Jomsom stations during the limited times when data were available from both, we were unable to determine whether this pattern persisted throughout the year. For this reason, our only option is to follow the findings of Egger et al. (2000) and assign the wind speed at JSM_2 to the entire flux plane.".

Line 440: please provide formula for this calculation. Which is the usefulness of calculating this values of mass exported per day? Flux is defined as "kg/m2·s": "kg /day" is not a flux! Please revise.
We have now provided a description of our flux calculations (including the formula) in the methods section. We have now also corrected our values and units as pointed out by the reviewer.

Line 570: so, which is the outcome of this comparison? Table 1: Are std.dev or 95th confidence levels reported?
We have removed this text. We have clarified the table caption to indicate that the reported values are mean and standard deviation.

Figure 7: what the whiskers and bars represent?
The figure caption has been edited to include a more detailed description of the what the

whiskers and bars represent. The revised figure caption now reads: "Figure 7. (a) BC concentration distribution with down-valley (Dv) and up-valley (Uv) flows in Jomsom, (b) calculated Dv and Uv flux for each season, (c) Net daily flux per season. The dotted line is panel c marks 0 mg m$^{-2}$ day$^{-1}$. The red line represents 50$^{th}$ percentile, the edge of the box 25$^{th}$ and 75$^{th}$ percentile while the whiskers show maximum and minimum values.".

Supplementary Table 2: no measurement units are reported. Please adopt a nomenclature compliant with Figure 7
The table description now includes abbreviations consistent with Figure 7.

Supplementary Table 3 . this should be moved to the main manuscript! For each episode provide information about O3: when is it above the 90th percentile? When "cloud cover" is reported, did you mean that source identification was not possible? Please explain the manuscript text…
We have moved Supplementary Table 3 to the main text (Table 3). We removed MODIS imagery data as it was not comprehensive.

Reviewer #2

The present manuscript explores air quality data from a remote valley location in the Himalayas and tries to suggest that deep Himalayan valleys may act as important export routes for pollution from the Indo-Gangetic Plain (IGP) to the Tibetan Plateau (TP). The authors have addressed several comments given for the previous version of the manuscript and added some additional analysis. However, they fail to dissolve many of the major remarks made previously and the presented results still do not indisputably support the conclusions drawn by the authors. In addition, the technical quality of the manuscript has not improved and does not meet ACP standards (language, presentation of results). Therefore, I would suggest to reject the manuscript in its current for and encourage the authors to take the extensive comments made by reviewer 1 more seriously and carefully rework the manuscript for a novel submission.

We thank the reviewer for his/her continued and valuable feedback with our manuscript. We have made every effort to thoroughly address the remaining issues raised by Reviewer #1 and have made a global and thorough revision of the text's structure and grammar. As the primary goal of this manuscript is observational, we have also exercised greater caution in presenting the more speculative and preliminary aspects (i.e., flux estimates) of our work and have eliminated an examination of pollutant source regions. Taken together, these changes as well as the one's informed by the remaining comments of Reviewer #2 have helped to greatly strengthen our manuscript. Our responses to the reviewer's specific comments can be found below.

Specifically, I have three major concerns with the present study that would need to be addressed more adequately before the manuscript could be considered for publication.

Local influences

The local environment of the site and how local emissions may influence the present results and conclusions needs way more attention. A quick look on google Earth revealed that the town of Jomsom is not that small at all and that there is even a small airfield. A map (or googleEarth image) to clarify the site location relative to local pollution sources would be helpful to exclude any bias of the measurements due to local contamination. It would be possible to use the local wind observations to eliminate phases when local emissions directly influence the measurements. As the authors themselves state, the observed morning peak in BC is most likely due to local emissions from biofuel burning. The same might be true for the evening peak, but is never considered. Next to the local emissions also emissions further down the valley should be carefully looked at before concluding that all the observed BC is advected from the IGP rather than from the valley itself.

We have added text to better justify that the evening peak is dominated by non-local sources. The added text now reads: "Further, the Jomsom station measuring BC and $O_3$ (JSM_STA) is located more than 100 m above the valley floor where the village of Jomsom sits. As such, we do not expect that local evening emissions would reach the stations at the cessation of up-valley flows and that evening drainage flows, following the valley floor would remove these local evening emissions down-valley.".

Flux calculation

In the revised manuscript the authors present an estimate of the BC flux through the valley. Although it is never stated how this flux was calculated, I assume the authors multiplied the local wind speed and concentration measurements to derive a flux. This is really oversimplifying the transport situation in a complex alpine valley! At least vertical profiles of wind speed and concentrations would be needed to derive a robust flux estimate, two-dimensional transects would be even better.

Especially the nighttime flux at the near-surface site cannot be seen as representative for the whole valley atmosphere, since the surface layer often decouples from the flow above and considerable drainage flows may exist above. The idea to present a flux estimation to corroborate their conclusion of net pollution export through the valley is honourable but the present database simply does not allow for a robust estimate of a flux.
We agree with the reviewer that our flux estimate is preliminary and coarse. While the flux estimate is not central to our study, the estimate that we present is meant as a rough, initial, order-of-magnitude assessment. We have now included a description and formula for our flux calculations (as well as the assumptions that informed our estimate) in the methods section. Having clearly listed these limitations and assumptions in the text, if the reviewer or editor still has objections to this calculation, we can remove this aspect of the paper if necessary. The added text now reads: "Up-valley and down-valley BC fluxes were calculated separately

and determined using wind direction measurements. Instantaneous flux at time $t$ was calculated as:

$$j_t = BC_t ws_t$$

where $BC_t$ is the BC concentration (mg BC m$^{-3}$) at time $t$ and $ws_t$ is the wind speed (m hr$^{-1}$) at time $t$. From this, net daily mass transport ($M_d$) for day $d$ was estimated as:

$$M_d = A \left( \sum \left( \frac{1}{6} j_{t,up} \right) - \sum \left( \frac{1}{6} j_{t,down} \right) \right)$$

where $A$ is the cross-sectional area of the valley at Jomsom ($1.41 \times 10^6$ m$^2$), $j_{t,up}$ and $j_{t,down}$ are the instantaneous up-valley and down-valley fluxes, respectively, occurring during day $d$, and the factor of 1/6 is the length of each time step in hours (i.e., 10 minutes divided by 60 minutes). The cross-sectional area was estimated as a trapezoid with a height of 800 m (the difference in elevation between the two Jomsom stations) and cross-valley distances of 800 m (at JSM_1) and 2720 m (at JSM_2). The average of these two cross-valley distances (1760 m) multiplied by the height (800 m) yields the cross-sectional area of $1.41 \times 10^6$ m$^2$.

If we assume that (1) the polluted boundary layer within the valley at Jomsom is 800 m deep (i.e., the approximate elevational difference between the two AWS sites at Jomsom), (2) BC within the polluted boundary layer is well mixed, and (3) wind velocities do not vary significantly with altitude through the polluted layer, the mass flux BC through a vertical plane across the valley can be estimated. Supplementary Figure 1 shows that JSM_2 is well within the polluted haze layer during daytime/upvalley flows and that some BC is almost certainly transported above 800 m elevation. In this way we ensure that our estimates are conservative. The long lifetime of particulate BC against deposition (several days to a week or more) coupled with turbulent flow within the valley supports the assumption that BC is well mixed. In addition, Egger et al. (2000) used theodolite measurements to demonstrate uniform wind speeds within the bottom 1000m above the KGV floor at Jomsom and other locations. While we observed differences in wind speed of ~5 m s$^{-1}$ between the two Jomsom stations during the limited times when data were available from both, we were unable to determine whether this pattern persisted throughout the year. These data limitations also prevented us from a more in-depth assessment of potential nighttime decoupling. For these reasons, our only option was to follow the findings of Egger et al. (2000) and assign the wind speed at JSM_2 to the entire flux plane.".

Trajectories

The presented analysis of trajectories and fire hot spots seems a bit arbitrary. The description of the back-trajectories is completely insufficient, both in terms of how they were calculated as well as how they are interpreted. The authors need to explain where and when the trajectories were initialized and why. Apparently the underlying meteorological data used for their calculation was insufficient to include the up-valley flow. So what was the meteorology used? The authors need to argue why it would be sufficient to include the transport from the

source regions to the valley mouth in any detail if model resolution is insufficient to cover the valley flow. The connection with the sources is too qualitative and vague to be used as evidence. The shown MODIS fire pixels can only be seen as an indication that there were some fires in the region, but they don't tell us anything about the absolute emission strength nor the relative emission strength during the whole observation period. More quantitative products of fire emissions are freely available and should be looked at (e.g., GFED, CAMS GFAS).

We have eliminated any assessment of pollution source. To do a thorough assessment would be beyond the scope of this paper. As such, we have included recommendations that future studies should examine this aspect, but now do not make any attempt to characterize pollution sources here.

[revised manuscript text omitted]
., 2012). Other studies show that pollutants have the potential to reach not only the foothills of the Himalaya but also higher elevations (Bonasoni et al., 2010; Decesari et al., 2010; Marinoni et al., 2010).Studies in the Himalayan region have looked into the transport of pollutants across the mountains (Lüthi et al., 2015) and the contribution of region sources like open fires to the occurrence of ABC (Ramanathan et al., 2005; Ramanathan and Crutzen, 2003). In addition, studies have shown the obstruction of flow caused by the high Himalaya which intensifies the effect of pollution over the IGP that are visible in satellite imagery especially during pre-monsoon seasons (Singh et al., 2004; Dey and Di Girolamo, 2010; Gautam et al., 2011).Other studies show that pollutants have the potential to reach not only the foothills of the Himalaya but also higher elevations (Bonasoni et al., 2010; Decesari et al., 2010; Marinoni et al., 2010). Though there is evidence of existence of similar source pollutants, both regional and local, in the foothills and the higher altitude sites (Raatikainen et al., 2017; Raatikainen et al., 2014; Srivastava et al., 2012,), observational evidence of Himalayan valleys as mechanisms and possible pathways facilitating such transport via Himalayan valleys is lacking.~~ Here we present 2.5 years of measurements of BC, $O_3$, and associated meteorological data from one of the deepest trans-Himalayan valleys, the Kali-Gandaki Valley. We examine seasonal and diurnal patterns of BC and $O_3$, investigate potential episodes of enhanced pollution transport up-valley, and make a preliminary estimate of BC mass transport. In doing so, we seek to provide the first

observational evidence of trans-Himalayan valleys acting as conduits for pollution transport from the IGP to the higher Himalaya . illustrates the importance of  the role of the wind system within a deep Himalayan valley in transporting pollutants from the IGP to the high mountains.

[revised manuscript text omitted]

 For example, the Nepal Climate Observatory-Pyramid (NCO-P) station at the 5079 m asl in the Himalaya has also shown high seasonal differences for BC  between pre-monsoon (0.444 (±0.443) µ·g BC m$^{-3}$; 61 (±9) ppbv O₃)  during pre-monsoon and monsoon (0.064 (±0.101) µ·g m$^{-3}$; 39 (±10) ppbv O₃) during monsoon  (Cristofanelli et al., 2010, Marinoni et al., 2013) (Table 1). Our results indicate a lagged peak  within the KGV and presumably other deep Himalayan valleys,

**3.2 Diurnal variability in BC and O₃**

[revised manuscript text omitted]

**3.3 Evolution of local wind system in the KGV**

Aning of is essential for analyzing pollution transport through mountain valleysMeasurements from JSM_2 show the diurnal evolution of wind in each season. All data collected were used to analyze the diurnal pattern of wind at Jomsom (Fig. 6). The Wwind roses illustrate the temporal evolution of up- and down-valley flows at JSM_2 for each season (Fig. 6). At JSM_2, up-valley flows are southwesterly and dominant during daytime, with peak velocities above 15 m s$^{-1}$ between about 0900 LT to 1800 LT. Wind velocities decreased substantially after 1800 LT, with variable wind direction until midnight, followed by then northeasterly winds are common (during pre- and post-monsoon seasons (Fig. 6a and 6c)). The wind patterns during monsoon appearwas strongly influenced by the monsoon anticyclone;, this observation is in agreement with wind direction measurements from other Himalayan valleys (Bonasoni et al., 2010; Ueno et al., 2008) (Fig. 6b). Although wind velocities at JSM_2 varied somewhat over the year, non-monsoon months exhibited similar diurnal patterns that evolved seasonally as a function of sunrise and sunset (Fig. 6), characteristics which were exhibited unlike in the monsoon season. As discussed below, this alternating pattern in wind direction from strong daytime flows to weak nighttime flows during dry months results in a net transport of pollutants up the valley.

Theodolite observations at different locations along the KGV in 1998 show minor shifts – less than 45° – in wind direction and less than 2 m s$^{-1}$ in wind speed in the lower 1000 m above the surface during daytime (Egger et.al 2000). Based on the average daytime wind speed, the valley can be partitioned into three regions: the entrance, core, and exit (average wind speeds range from 5 to 10 m s$^{-1}$, 8 to 18 m s$^{-1}$, and less than 5 m s$^{-1}$, respectively) (Egger et al 2000). The strongest winds within the core region are most prevalent in the lower 1000 to 1500 m of the boundary layer within the valley (Egger et al, 2000; Zängl et al., 2000, Egger et al., 2002). Our measurements at the four AWS stations on the valley floor illustrate the evolution of surface wind velocities along the length of the KGV (Supplementary Figure 1). In general, wind speeds along the valley floor were strongest within the core of the valley at MPH,

**3.43.1 Local winds as drivers of BC and O$_3$ transport in the KGV**

Figure 4 6 shows the time series of BC and O$_3$ during individual months or in each seasons (April/pre-monsoon, August/monsoon, November/post-monsoon). For pre-monsoonApril, BC concentrations peaked at 0700 LST (Fig. 46) when wind velocities were low (Fig. 46). This peak occurred about an hour later during the post-monsoon periodin November, with all seasonsboth periods experiencing a decrease in BC over the rest of the morning as wind speeds increased and diluted local emissions (Fig. 45). Dilution of local emissions associated from increasing wind speed likely contributes to decreasing BC concentration during late morning. Thereafter, BC concentrations increased over the afternoon and early night, reaching secondary peaks near midnight LST during in the pre-monsoonApril and several hours earlier during other periods in November and August (Fig. 4 and 56). Distinct morning and afternoon peaks in BC concentration are seen in the post-monsoon season when the up-valley wind speeds are relatively calmer weaker than in pre-monsoon season (Fig. 5 4 and 6). The bBimodal diurnal distribution of diurnal BC concentration in Jomsom is similar to that observed in Kathmandu as reported by (Putero et al. (2015) butand unlike a singular late afternoon/evening peak in the late afternoon/evening inseen at high elevation sites (Bonasoni et al., 2010) where similarduring nonmonsoonal seasons. This illustrates that deep Himalayan valleys are susceptible to diurnal pollution similar to that of urban areascities like Kathmandu. The morning peak in BC is most likely due to diurnal peaks were observed (Bonasoni et al., 2010; Hindman et al., 2002; Hegde et al., 2007 ). local pollutants (from household and morning aircraft traffic) in Jomsom and settlements downwind of Jomsom,. Wwhile the afternoon peak is likely primarily associated with long range in addition to local pollutants. At the same time, Simultaneously, $O_3$zone exhibits a distinct minimum in the early morning with concentrations increasing towards an early afternoon peak – occurring well before BC's afternoon peak. However, mixing ratios increase during the morning, peak in the early afternoon well before that of BC and decrease over night. The $O_3$ minimuma in the morning further supports that the morning BC peak originates from corroborates the role of local sources,pollutants responsible for morning peak as since $O_3$ is only formed downwind of pollution sources. Further, the Jomsom station measuring BC and $O_3$ (JSM_STA) is located more than 100 m above the valley floor where the village of Jomsom sits. As such, we do not expect that local evening emissions would reach the stations at the cessation of up-valley flows and that evening drainage flows, following the valley floor would remove these local evening emissions down-valley.

Percentage distributions of up-valley and down-valley BC concentrations, fluxes, and net daily fluxes per unit cross section are depicted in Figure 7 and summarized in Supplementary Table 21. Up-valley (southwesterly) flows are defined as between 35° and 55° while down-valley (northeasterly) flows include data between 215° and 235°. Date for all days for which complete data were available over entire 24-hour periods were binned by season. The statistical significance of differences between up-valley and down-valley flow conditions during different seasons were evaluated using the non-parametric Kruskal Wallis and Mann-Whitney tests. The distribution ofdifferences in BC concentrations between up-valley and down-valley flows are quite similar yet statistically significant for all seasons, with higher down-valley concentrations (Figure 7a). However, because wind velocities were relatively higher during up-valley daytime flow and the durations of up-valley were modestly longer than those of down-valley flow, the corresponding up-valley fluxes of BC during daytime were markedly and significantly greater than down-valley fluxes during all seasons (Fig. 7b). These results suggest an oscillatory movement of polluted air within the valley, where polluted air masses are pushed up-valley during daytime and retreat a shorter distance during nighttime. These Ddifferences between up-valley andversus down-valley fluxes yielded significant net dailypositive up-valley fluxes of BC during all seasons (Fig. 7c). Because heating would have driven growth of the boundary layer and thus greater ventilation and dilution of pollutants during daytime relative to night, we infer that the calculated differences in up-valley versus down-valley fluxes correspond to lower limits for net BC fluxes.

450 Positive up-valley fluxes are consistent with an "alpine pumping" mechanism in the Himalayan valleys and thereby support the hypothesis that these valleys are important pathways for pollution transport. If we assume that (1) the polluted boundary layer within the valley at Jomsom is 800 m deep (i.e., the approximate elevational difference between the two AWS sites at Jomsom), (2) BC within the polluted boundary layer is well mixed, and (3) wind velocities do not vary significantly with altitude through the

455 polluted layer, the mass flux BC through a vertical plane across the valley can be estimated.  Because . Supplementary Figure 2 shows that during daytime JSM_2 is well within the polluted haze layer during daytime/ upvalley flows and some BC is almost certainly transported above 800 m elevation where wind speed is about 5 m s$^{-1}$ higher, thus this approach yields a conservative estimate.  The long lifetime of particulate BC against deposition (several days to a week or more) coupled with turbulent flow

460 within the valley supports the assumption that BC is well mixed.  As noted previously, wind velocities measured at JSM_1 (2800 m) and JSM_2 (3700 m) were similar suggesting minimal variability through the lower 800 m depth of the valley.  Extrapolation of the net daily BC flux per unit area to the cross-sectional plane of the valley at Jomsom (1.62 km$^2$) yields  In addition, we estimate substantial net daily mass transportfluxes amount of BC through the up the valley during the pre-monsoon season: of 1.05

465 kg day$^{-1}$ (based on the average daily net daily flux) and 0.72 kg day$^{-1}$ (based on the median daily net daily flux). While preliminary, these Such estimates provide the first useful semi-quantitative constraints on mass fluxes transport of BC from the IGP to the high Himalaya through deep valleys and, more generally, on the regional cycling of BC over southern Asia.

470 **3.5 Evidence of regional transport episodes in valley concentration**

Along with the regular diurnal and seasonal variability driven by local winds as described above, we also observed anomalous periods – lasting when for several days to more than a week – during which BC concentrations of both BC and O₃ were significantly greater than the 90th percentile for corresponding annual averages (Table 3). (Supplementary Table 3). These extended periods of high BC

475 and O₃ at JSM_STA are evidence of large-scale transport from the IGP to the foothills in conjunction with local valley winds (Fig. 8). We identified three common patterns in the BC profile at Jomsom during the transport episodes. For most Most of the transport episodes were associated with observational evidence from satellite imagery of emission sources within the region including haze over IGP, agricultural and biomass burning in Punjab regions of India and Pakistan, and forest fires in the

480 foothills of the Himalaya in northern India or southern Nepal (Supplement Table 3). Pollutant source location could not be identified at times due to cloud cover. The 90th percentile for each year 2013, 2014

 Elevated $O_3$ concentrations did not always accompany these long-duration periods of high BC concentrations , above $90^{th}$ percentile. Different atmospheric lifetimes, chemical reactivity, source location, and sinks of BC and $O_3$ all may contribute to these differences in BC and $O_3$ concentrations detected in the valley. Table 3 reports the number of days in which $O_3$ concentration was above $90^{th}$ percentile within each episode.

We partitioned these episodes into three characteristic patterns based on the relative variability of BC. Pattern A was characterized as a fluctuating daily maximum in BC with peaks that repeatedly exceeded the $90^{th}$ percentile but with daily minima below the $90^{th}$ percentile (Fig. 8[Ia]). Pattern B was characterized by a steady buildup of BC concentration  over the period of the regional transport episode, with peak BC concentrations over the $90^{th}$ percentile but without a  low daily minimum (Figure 8b).  Pattern C exhibited a combination of  both Ppatterns A and B during a single regional episode (Figure 8c). A total of 34 regional episodes were identified from January 2013 through June 2015, 47% of which were categorized as Ppattern A, 32% as Pattern B, and 21% as pPattern C. The wind speeds at Jomsom during these transport episodes exhibited diurnal variability similar to those during other periods (Fig. 8 [II]).

During the regional transport period in November 2014 (Pattern A), average daily BC concentration was 1.29 µg m⁻³ which is over the $75^{th}$ percentile (0.88 µg m⁻³) of the BC concentration for the measurement duration. The maximum daily concentration during the period was 3.04 µg m⁻³. However, the corresponding average $O_3$ was only 28.07 ppbv, slightly below the average (29.48 ppbv) for the entire data set (Figure 8 [Ia]).  The mean BC concentration during pattern B, one example of which occurred in May 2014, was 1.77 µg m⁻³ (Figure 8[Ib]). It was above the $90^{th}$ percentile (1.49 µg m⁻³) for entire measurement period while $O_3$ concentrations were at 49.71 ppbv, slightly below the $90^{th}$ percentile (52.9 ppbv).

valley flows compared to pattern A (Fig. 8 [IIb]). One of the Pattern C- type of transport episodes was identified in May 2013, when the average concentration was well above the 90th percentile for both BC (2.09 µg m$^{-3}$) and O$_3$ (57.49 ppbv) (Fig. 8 [Ic]). Diurnal wind patterns were similar to those of pattern B with extended duration of up-valley flows (Fig. 8 [IIc]). The diurnal wind pattern in the KGV was conserved during the Pattern A example, but a longer period of up-valley flows occurred during the examples for Patterns B and C (Fig. 8).

We initiated the Hybrid Single-Particle Lagrangian Integrated Trajectory model (HYSPLIT), developed by NOAA's Air Resources Laboratory, from 300 m, 500 m and 1000 m for 72 hours runtime for each of the example episodes (Stein et al., 2015). The back trajectories were initiated at the mouth of the valley on the day when fire events were the most visible on MODIS imagery within each example episodes in Figure 8. The results though mostly qualitative show that regional episodes occurred when extensive haze or fire events were detected in the Indo-Gangetic plains (Supplementary Figure 2). The MODIS fire data and HYSPLIT back trajectories showed extensive haze and fire events during this period of regional transport (Figure 9). We infer that transport of BC emitted from agricultural burning and wildfires during this period contributed to the high concentration measured with the KGV.

The mean BC concentration during pattern B, one example of which occurred in May 2014, was 1.77 µg m$^{-3}$ (Figure 8[Ib]). It was above the 90th percentile (1.49 µg m$^{-3}$) for entire measurement period while O$_3$ concentrations were at 49.71 ppbv, slightly below the 90th percentile (52.9 ppbv). The wind pattern in the KGV exhibited diurnal flow patterns but with longer period of up-valley flows compared to pattern A (Fig. 8 [IIb]). During this period MODIS imagery and HYSPLIT back trajectories revealed widespread burning in the Himalayan foothills and the Punjab region of India (Figure 9).

One of the Pattern C- type of transport episodes was identified in May 2013, when the average concentration was well above the 90th percentile for both BC (2.09 µg m$^{-3}$) and O$_3$ (57.49 ppbv) (Fig. 8 [IIc]). Diurnal wind patterns were similar to those of pattern B with extended duration of up-valley flows (Fig. 8 [IIc]). MODIS and HYSPLIT back trajectories revealed extensive agricultural burning in the Punjab region of India and the southern plains of Nepal during this period (Figure 9).

3.5 Pollution in the Higher Himalaya

Average BC concentrations in European cities including Barcelona, Lugano and London range between

1.7 to 1.9 µg m⁻³ (Reche et al., 2011). National Ambient Air Quality Standards (NAAQS) for the United States for O₃ is 70 ppbv (8-hour maximum average). During regional transport episodes, BC and O₃ levels in the remote site in Jomsom occasionally exceeded the above levels for both BC and O₃. BC and O₃ concentrations suggest that the extensive haze layer over the IGP during pre-monsoon along with increased fire activity in the IGP and Himalayan foothills results in an increased flux of pollutants into the Himalayan valleys (Fig. 7 and 8). The pollutants are transported by the local wind system, specifically the up valley winds that are dominant throughout the day. However, larger synoptic patterns are responsible for transport of pollutants from regional sources to the Himalayan foothills and the entrance region of valleys like the KGV.

The results obtained at JSM_STA show similar seasonal pattern as NCO-P CNR (Bonasoni et al., 2010) with higher concentration of BC and O₃ in the pre-monsoon season and lowest during the monsoon. However, the magnitude of concentration differ for both BC and O₃ at the two sites (Table 1). Our results from JSM_STA found BC concentrations in the KGV are twice as high as those measured at NCO-P CNR during pre-monsoon season while O₃ concentration at NCO-P CNR were twice as high as JSM_STA. At JSM_STA during the post monsoon season, BC concentration are comparative to pre-monsoon season and 5 times higher than that at NCO-P CNR while O₃ concentrations remain higher at NCO-P CNR. The NCO-P CNR is at a higher elevation than Jomsom and can be affected by frequent stratospheric intrusions thus resulting in higher O₃ concentrations than that at Jomsom (Cristofanelli et al., 2010). Alternatively, BC is significantly higher in Jomsom for all seasons when compared to NCO-P CNR.

**4. Conclusion**

This study provides new in-situ observational evidence of the role of a major Himalayan valley as an important pathway for transporting air pollutants from the IGP to the higher Himalaya. We found that:

Concentrations of BC and O₃ in the KGV exhibited systematic diurnal and seasonal variability. The diurnal pattern of BC concentrations during the pre- and post-monsoon seasons were modulated by the pulsed nature of up-valley and down-valley flows. Seasonally, pre-monsoon BC concentrations were higher than in post-monsoon season.

We also found that The morning and afternoon peaks in the post-monsoon season were more pronounced than those of pre-monsoon season, likely due to the relatively lower wind speeds during post-monsoon.

When compared to a high elevation site, NCO-P CNR in the Himalaya, JSM_STA consistently showed higher BC concentrations for all seasons whereas the corresponding O₃ concentrations were higher at NCO-P CNR.

Significant positive up-valley fluxes of BC were measured during all seasons and preliminary flux estimates (which require a more robust estimate in future work) show the efficiency and magnitude of pollutant transport up the valley.

During episodes of regional pollution over the IGP, relatively higher concentrations of BC and O₃ were also measured in the KGV.

The frequency and magnitude of pollution events highlighted in the paper need to be studied for a longer period in order to understand the associated interannual variability. The flux measurement provided shows the efficiency of pollutant transport through the valley. However, further studies are required for a robust quantification of the fluxes. Further In addition, future work should focus on studies are needed to understanding the vertical and horizontal distribution of particulate matter and ozone in the Himalayan region, and their impacts on the radiative budget, the ASM and regional climate. Investigations using sondes, LiDar and air-borne measurements could help characterize the stratification of the vertical air masses.

**Acknowledgments**

We would like to acknowledge our field assistant in Nepal, Buddhi Lamichhane who helped us in various stages of the study, as well as the logistic and administrative support and internet at the Jomsom station provided by Nepal Wireless. Financial support was provided by the National Aeronautics and Space Administration NNX12AC60G, and additional field support was provided by ICIMOD's Atmosphere Initiative. The authors are very thankful for comments from William Keene and Jennie Moody.

[revised manuscript text omitted]

---

## Author Response (AR3)

*Dear Dr. Stone,*

*We would like to thank the referees for their constructive comments, which have helped us to further improve the clarity and quality of the manuscript. Specifically, we have: 1) performed a substantial formatting and re-working of the manuscript tables and figures, 2) clarified the methods related the estimates of net daily flux, and 3) included additional cautionary statements regarding the conclusions that can (and cannot) be drawn from our study and findings. The responses to the referees are included below. Each referee comment is followed by our response to that comment (in italics). With these improvements incorporated into our manuscript, we now feel that our manuscript is now suitable for publication in ACP.*

*Thank you for your continued consideration,*

*Shradda Dhungel\*, Bhogendra Kathayat, Khadak Mahata, and Arnico Panday*

*\*shradda@virginia.edu*

**Referee #1**

This is my third review of this manuscript. I can say that the authors made a good work with this last revision. The manuscript presentation is significantly improved and the results are discussed in an appropriate and balanced way. Thus, I can recommend pubbication but some technical corrections must be considered.

Line 14-16: something appear to be missed in this sentence. Please, check. Maybe: "...to the higher Himalayas. Nevertheless, observational evidences of the transport of polluted air-masses across Himalayas has been lacking to date."

> *We have clarified the text to read: "Models and satellite imagery suggest that strong wind systems within deep Himalayan valleys are major pathways by which pollutants from the IGP are transported to the higher Himalaya. However, observational evidence of the transport of polluted air masses through Himalayan valleys has been lacking to date."*

Line 22-23, page 3: I do not agree: e.g. Bonasoni et al. (2010) showed clearly the effect of thermal wind circulation on the transport of pollutants along an Himalayan valley.

> *We have added Bonasoni et al. 2010 reference.*

Line 33, Page 3: too much "previously"!

> *We have removed the latter use of 'previously'.*

Line 8-9, page 5: please check the wind sectors (are they inverted?)

> *We have made the correction as indicated by the referee.*

Line 20-27, page 5: the formula is not clear to me. Why did you express wind speed in m/hr (and not in m/s)? Is not clear to me, the reason for using the factor 1/6. On which basis did you perfom the summation? I suppose over time but please, explicitate it! Please, revise this formula and correct it accordingly....

*We have now included a Δt term (equal to 600 seconds) within the formula and ensured that the our time units are in seconds that we explicitly state how the summation was performed. The modified text now reads: "Up-valley and down-valley BC fluxes were calculated separately and determined using wind direction measurements. Instantaneous flux at time t was calculated as:*

$$j_t = BC_t ws_t$$

*where $BC_t$ is the BC concentration (mg BC $m^{-3}$) at time t and $ws_t$ is the wind speed (m $s^{-1}$) at time t. From this, net daily mass transport ($M_d$) for day d was estimated as:*

$$M_d = A\left(\sum\left(j_{t,up}\,\Delta t\right) - \sum\left(j_{t,down}\,\Delta t\right)\right)$$

*where A is the cross-sectional area of the valley at Jomsom (1.41x$10^6$ $m^2$), $j_{t,up}$ and $j_{t,down}$ are the instantaneous up-valley and down-valley fluxes, respectively, occurring during day d, and Δt is the length of each time step in seconds (i.e., 600 seconds). The cross-sectional area was estimated as a trapezoid with a height of 800 m (the difference in elevation between the two Jomsom stations) and cross-valley distances of 800 m (at JSM_1) and 2720 m (at JSM_2). The average of these two cross-valley distances (1760 m) multiplied by the height (800 m) yields the cross-sectional area of 1.41x$10^6$ $m^2$. Thus, net daily flux was calculated as the difference between the summation of instantaneous up-valley fluxes for day d and the summation of instantaneous down-valley fluxes for day d.".*

Line 29, page 6: Fig 5b? It is correct?

*We have corrected the figure citation from Fig. 5b to Fig. 3b.*

Line 30, page 7: Maybe better Fig.6

*We have corrected the figure citation from Fig.5 to Fig. 6.*

Line 20, pag. 8: "(Fig. 4 and 6)"

*We have corrected the text as advised.*

Pag 10, line 5: only 1 number after decimal for O3.

*We have corrected the text as advised.*

**Referee #2**

This is my first time to review this MS. I totally agree with comments from those previous referees. I find that numbers of problems still existed in the new version of the MS. I suggest the MS a major revision and the following are some more suggestions to the authors.

P3 Line 3 reference of (Kopacz et al., 2011) should be replaced by (Zhang et al., 2015) because the results of the former overlooked the potential contribution of local Tibet, which is out of date.

*We now also reference Zhang et al., 2015.*

2.      P3 Line 31 "no pollution data from Jomsom have been previously reported previously". The authors overlooked one just published work last year that has done 14C study of BC at Jomsom, which showed the influence of BC from South Asia to the Himalaya and the local biomass burning sourced BC to valleys of the Himalaya (Li et al., 2016). In addition, please add (Chen et al., 2017) and (Luthi et al., 2015), which are also two related article of this study.

*We have now included all three references as suggested.*

3.      I am sorry to say that the figures and tables are not typical for both ACP and other scientific journals, which have been pointed out by other referees. However, the authors make little changes and still have lots of problems. For instance, you should at least add longitude and latitude to the figure 1 so that the readers know where your research site located. You also need to point out where is South Asia, where is the Tibetan Plateau at this figure.

*We have improved the clarity of the figure and its caption. We have included an inset in Panel (a) to show the location of the NASA image. We have also indicated the location of the Kali Gandaki Valley, the Indo-Gangetic Plain, the Himalaya, and the Tibetan Plateau. The updated caption now reads: "Figure 1. (a) NASA Worldview image from November 4th , 2014 depicting thick haze intruding the Himalayan foothills with red arrow over the KGV. Inset map in the top-right corner of panel (a) shows the location of the Worldview image. Dashed box shows the location of panel (b). (b) Expanded scale of the KGV showing station locations of Lete (LET_AWS) near the entrance of the valley (yellow star); Marpha (MPH-AWS) in the core region (gray star); the Jomsom (JSM_STA) sampling station for BC and $O_3$ and the two associated AWS sites (JSM_1 and JSM_2) in the core region (orange star); and Eklobhatti (EKL_AWS) near the exit (blue star). (c) The atmospheric observatory at Jomsom (JSM_STA) (28.87° N, 83.73° E, 2900 m asl). (d) Valley cross-sectional elevation profile at the station locations.".*

Figure 2 looked a little strange and is useless. Instead, the words that point out four sites should be moved to right photo of figure 1. The table of Figure 2 should be moved to supporting information.

*We have removed Figure 2. The site information previously included therein has been moved to the figure caption of Figure 1, and the table has been moved to supporting information as the new Supplementary Table 1.*

The expression of tables is not typical, which should be "Three line table". Please change the format of tables.

*We have updated all of the tables to be "three line tables".*

Finally and honestly, the English of the article is not fluent, please improve it by expert English editor.

*The grammar and flow of the entire text has been thoroughly checked by multiple native English speakers. We leave it to the Editor to decide.*

**Referee #3**

This is my third review of the manuscript. In comparison to my previous review the authors have partly addressed my three main points of concern although not necessarily in as much detail as I had hoped for. However, they have removed some more speculative parts of the discussion of the pollution transport towards the valley, so that this part is not hindering publication anymore. Nevertheless, I still have some further comments concerning the last additions and I make some additional recommendations concerning the presentation of figures and tables, which in my view are still not up to ACP standards. Once these issues are addressed publication in ACP could be considered.

Local influences

The authors have replied to my concerns about local pollution episodes by adding a sentence on the use of the local air strip which suggests that this will only have minor influences. However, the authors did not follow the suggestion of further screening their data based on local wind direction and location of sources, but argue that the vertical distance to the town is large enough to minimise the influence. Without further analysis and data this remains hard to judge, but there seems to be sufficient mention of local sources now to create the required cautiousness when interpreting the results.

*We agree that it is important to account for local sources and agree that the existing text conveys sufficient caution to the reader. The text in Section 3.4 not only cautions the reader that not all of the BC that we measure necessarily originates from the IGP but also explicitly lists the active local sources that may partially contribute to the diurnal patterns that we observe.*

Flux calculation

The authors added a whole section explaining the calculation of net upvalley BC fluxes and also explain the limitations of their approach. One of the main assumptions of the approach is that of constant depth of the upvalley or downvalley flows which is clear oversimplification of the flow especially during nighttime. The authors should add another sentence on the nighttime flow. It should be mentioned that their assumption is most likely an overestimation of the nighttime flow depth and, hence, nighttime downvalley flux. Therefore, their net upvalley flux is most likely an underestimation and at least the conclusion that there is net upvalles mass flux seems to be supported by the approach, whereas the absolute numbers should definitely taken with caution. I would also suggest a rewrite of the equation of the daily net mass transport. Instead of using 1/6 in front of each term, it would be more logical to just a multiplication with Delta t at the end of the equation. This would immediately make clear that we are looking at an integration over time. The value of Delta t can then be given in the text.

*We have modified the equation as suggest and included a statement indicating how our choice of polluted boundary layer depth likely affects our daytime and nighttime flux estimates. The added text now reads: "Supplementary Figure 1 shows that JSM_2 is well within the polluted haze layer during daytime/upvalley flows and that some BC is almost certainly transported above 800 m elevation above the valley floor. In addition, it is important to note that – by assuming a constant polluted boundary layer depth throughout the day – to a certain degree we likely underestimate up-valley flux during daytime and overestimate down-valley flux during nighttime. We therefore ensure that our estimates are conservative.".*

Trajectories

The previously criticised section on transport towards the valley mouth discussed by the analysis of air mass trajectories was removed from the manuscript. Therefore, my suggests for improving this kind of analysis were not followed, however, the manuscript does not draw any vague conclusions on this point anymore.

*While we agree with the referee that this would be important for better understanding the pollutant sources, we ultimately decided that such an analysis was beyond the scope of our study. Future work will investigate this in detail.*

Presentation of figures and tables

Some of the figures and tables still need improvement before publication in ACP.

*We have improved the formatting of all tables based on the suggestions of the other referees. Figure 1 has been improved based on the suggestions of one of the other referees. The previous Figure 2 has been removed, and the information from its table has been moved to the supplementary materials. The figure showing daily net flux (now Figure 6) has also been improved.*

Figure 2 should be split into a figure and a table! I think I mentioned this before in an earlier review ...

*We have moved the information from the table to the supplementary materials (now Supplementary Table 1). The image has been removed, and the information on station location and name has been included in Figure 1.*

Table 1 looks more like a figure than a table. Please check if this agrees with ACP guidelines for tables!

*We have made the previous Table 1 into a Figure (now Figure 2).*

Figure 3: Maybe wind-roses are just not the best way to present the wind statistics in the case of such a clear bidirectional flow. Maybe a pdf of wind speed for 3 different wind sectors would be much clearer. The figure right now consists mostly of white space and the interesting information makes up only a very small fraction of the plot and is hard to see.

*We prefer to keep wind roses for this figure, as we feel they adequately convey the changes in wind speed and direction both throughout the day and between seasons. To minimize the space demanded by the figure, we have moved the wind roses for the Monsoon and Post-Monsoon periods to the supplementary materials, keeping only the wind roses for the Pre-Monsoon season in the main text.*

Figure 7: In general the figure quality could be improved by not squeezing everything together in the vertical. Make the figure less wide, but add some vertical space between individual panels. I would also suggest to use a different y-axis scale for the monsoon period. Like it is presented now one cannot even tell the magnitude of the fluxes.

*We have improved Figure 7 as suggested.*

[revised manuscript text omitted]

---

## Author Response (AR4)

Comments to the co-editor:

Thank you for the comments to finalize the manuscript. We have made the suggested changes as shown below (in blue ink).

Thank you for your help throughout the process,

Shradda Dhungel

**Co-editor's comments**

Referee 1's comment on Line 22-23, page 3 of "I do not agree: e.g. Bonasoni et al. (2010) showed clearly the effect of thermal wind circulation on the transport of pollutants along a Himalayan valley…" pertains to the statement that "Such direct evidence of a Himalayan mountain valley wind system and its role in pollution transport has yet to be observed." Thus, adding the Bonasani et al. (2010) reference to the previous sentence does not yet address this reviewer comment. Please see section 4.1 in the referenced paper and revise the statement beginning with "Such direct evidence..." accordingly.

We have removed the reference from the preceding sentence and have modified the latter sentence. It now reads, "The resultant differences in temperature create gradients in pressure and density, which in turn drive the transport of air from the plains to higher elevations during the daytime (Reiter and Tang 1984, Whiteman and Bian, 1998; Egger et al., 2000). Such direct evidence of a Himalayan mountain valley wind system and its role in pollution transport has been observed at 5079 m a. s. l., particularly during non-monsoon seasons, in the Khumbu valley in Eastern Himalaya (Bonasoni et al., 2010). However, the transport patterns of pollutants at the habitable mid-altitudes within a trans-Himalayan valley have yet to be observed."

Referee 2's comment 3 was not yet addressed: "…you should at least add longitude and latitude to the figure 1 so that the readers know where your research site located."

We have added Lat/Lon to panel b of figure 1.

Regarding Referee 3's comment on "local influences" it should be clarified that the relative contributions of local and long range transport are not know. This could be done by adding "The contribution of long range transport relative to local sources of pollutants, however, is not known…"
before the sentence beginning "At the same time, O3…"

> We have clarified the sentence that relative distribution between local and long range pollutants

[revised manuscript text omitted]